# Response to Nodal morphogen gradient is determined by the kinetics of target gene induction

Julien Dubrulle[1]*, Benjamin M Jordan[2,3], Laila Akhmetova[1†], Jeffrey A Farrell[1†], Seok-Hyung Kim[4], Lilianna Solnica-Krezel[5,6], Alexander F Schier[1,7,8,9,10]*

[1]Department of Molecular and Cellular Biology, Harvard University, Cambridge, United States; [2]Department of Mathematics, College of Science and Engineering, University of Minnesota, Minneapolis, United States; [3]Department of Organismal and Evolutionary Biology, Harvard University, Cambridge, United States; [4]Division of Medicine, Medical University of South Carolina, Charleston, United States; [5]Department of Developmental Biology, Washington University School of Medicine, Saint Louis, United States; [6]Department of Medicine, Vanderbilt University Medical Center, Nashville, United States; [7]Harvard Stem Cell Institute, Harvard University, Cambridge, United States; [8]Broad Institute of MIT and Harvard, Cambridge, United States; [9]Center for Brain Science, Harvard University, Cambridge, United States; [10]Center for Systems Biology, Harvard University, Cambridge, United States

*For correspondence:
dubrullejulien@gmail.com (JD);
schier@fas.harvard.edu (AFS)

†These authors contributed
equally to this work

Competing interests: The authors declare that no competing interests exist.

**Abstract** Morphogen gradients expose cells to different signal concentrations and induce target genes with different ranges of expression. To determine how the Nodal morphogen gradient induces distinct gene expression patterns during zebrafish embryogenesis, we measured the activation dynamics of the signal transducer Smad2 and the expression kinetics of long- and short-range target genes. We found that threshold models based on ligand concentration are insufficient to predict the response of target genes. Instead, morphogen interpretation is shaped by the kinetics of target gene induction: the higher the rate of transcription and the earlier the onset of induction, the greater the spatial range of expression. Thus, the timing and magnitude of target gene expression can be used to modulate the range of expression and diversify the response to morphogen gradients.

## Introduction

The Nodal signaling pathway plays essential roles in animal development. Nodal signaling induces and patterns mesendoderm and establishes left-right asymmetry (*Conlon et al., 1994*; *Shen, 2007*; *Grande and Patel, 2009*; *Schier, 2009*; *Duboc et al., 2010*; *Shiratori and Hamada, 2014*). The Nodal signaling pathway regulates dozens of genes, ranging from transcription factors to cytoskeletal components, in order to pattern embryonic tissues (*Bennett et al., 2007*; *Liu et al., 2011*; *Fodor et al., 2013*). In embryonic stem cells, Nodal signaling is required for self-renewal as well as specification of endoderm and mesoderm (*James et al., 2005*; *Vallier et al., 2005*; *Schier, 2009*; *Oshimori and Fuchs, 2012*; *Chen et al., 2013*). Nodal signals can form concentration gradients and can act as morphogens (*Chen and Schier, 2001*; *Williams et al., 2004*; *Müller et al., 2012, 2013*; *Xu et al., 2014a*). It is unclear, however, how different Nodal concentrations induce different target genes and give rise to different cell types.

The classic morphogen threshold model postulates that Nodal signals are secreted from a source and form a concentration gradient that induces different fates in the target tissue according to local

**eLife digest** How a cell can tell where it is in a developing embryo has fascinated scientists for decades. The pioneering computer scientist and mathematical biologist Alan Turing was the first person to coin the term 'morphogen' to describe a protein that provides information about locations in the body. A morphogen is released from a group of cells (called the 'source') and as it moves away its activity (called the 'signal') declines gradually. Cells sense this signal gradient and use it to detect their position with respect to the source. Nodal is an important morphogen and is required to establish the correct identity of cells in the embryo; for example, it helps determine which cells should become a brain or heart or gut cell and so on.

The zebrafish is a widely used model to study animal development, in part because its embryos are transparent; this allows cells and proteins to be easily observed under a microscope. When Nodal acts on cells, another protein called Smad2 becomes activated, moves into the cell's nucleus, and then binds to specific genes. This triggers the expression of these genes, which are first copied into mRNA molecules via a process known as transcription and are then translated into proteins. The protein products of these targeted genes control cell identity and movement.

Several models have been proposed to explain how different concentrations of Nodal switch on the expression of different target genes; that is to say, to explain how a cell interprets the Nodal gradient. Dubrulle et al. have now measured factors that underlie how this gradient is interpreted. Individual cells in zebrafish embryos were tracked under a microscope, and Smad2 activation and gene expression were assessed. Dubrulle et al. found that, in contradiction to previous models, the amount of Nodal present on its own was insufficient to predict the target gene response. Instead, their analysis suggests that the size of each target gene's response depends on its rate of transcription and how quickly it is first expressed in response to Nodal.

These findings of Dubrulle et al. suggest that timing and transcription rate are important in determining the appropriate response to Nodal. Further work will be now needed to find out whether similar mechanisms regulate other processes that rely on the activity of morphogens.

ligand concentration (*Ashe and Briscoe, 2006*; *Barkai and Shilo, 2009*; *Rogers and Schier, 2011*). According to this model, high-threshold genes require high levels of Nodal signaling and thus are expressed close to the source (short-range genes), whereas low-threshold genes require lower levels of Nodal and are expressed at a greater distance from the source (long-range genes). Studies of mesendoderm patterning by Nodal in fish and frog have provided five lines of evidence that support the concentration threshold model. First, Nodal signals are produced locally starting at mid-blastula stages, and by the beginning of gastrulation, cells overlapping or close to the Nodal source express endodermal markers, while cells farther away express mesodermal genes (*Feldman et al., 1998*; *Sampath et al., 1998*; *Gritsman et al., 2000*; *Chen and Schier, 2001*; *Harvey and Smith, 2009*). Second, a gradient of activated Smad2, the principal transducer of the pathway, peaks at the Nodal source (*Faure et al., 2000*; *Yeo and Whitman, 2001*; *Harvey and Smith, 2009*) with high levels of activated Smad2 in endodermal progenitors and lower levels in mesodermal precursors. Third, reduction of Nodal signaling during blastula stages leads to the absence of endodermal fates but leaves most mesodermal fates intact (*Schier et al., 1997*; *Feldman et al., 1998*, *2000*; *Gritsman et al., 2000*; *Dougan et al., 2003*; *Hagos and Dougan, 2007*). Fourth, ubiquitous low concentrations of Nodal induce mesodermal markers, whereas high Nodal concentrations induce endodermal markers (*Gritsman et al., 2000*; *Thisse et al., 2000*; *Dougan et al., 2003*). Fifth, an ectopic source of Nodal can induce short- and long-range expression of endodermal and mesodermal markers, respectively (*Thisse et al., 2000*; *Chen and Schier, 2001*; *Williams et al., 2004*; *Müller et al., 2012*; *Xu et al., 2014a*). These observations suggest that different concentration thresholds induce different gene expression patterns.

In addition to the contribution of Nodal concentration to target gene induction, the timing of signaling affects Nodal interpretation. For example, the Nodal gradient is not static as signaling activity increases in range and amplitude between the initiation of Nodal expression and the onset of zebrafish gastrulation 2 hr later (*Harvey and Smith, 2009*; *Müller et al., 2012*). Moreover, delayed activation or premature inhibition of Nodal activity affects mesendoderm patterning (*Gritsman et al., 2000*;

*Dougan et al., 2003*; *Hagos and Dougan, 2007*). Two models have addressed how duration of exposure and changes in concentration contribute to Nodal interpretation. In the snapshot model, cells rapidly adapt their output to the increasing concentration of Nodal, regardless of the duration and history of exposure (*Rogers and Schier, 2011*). Indeed, increases in activated Smad2 levels are accompanied by an expansion of target gene expression domains (*Harvey and Smith, 2009*). In this model, the only role of time is to allow the gradient to expand and reach the thresholds that trigger the expression of short- and long-range genes (*Harvey and Smith, 2009*). The alternative 'cumulative dose' or 'integration' model postulates that the duration of Nodal signaling plays a critical role in Nodal interpretation. Cells adopt progressively more marginal fates with increasing duration of exposure to Nodal (*Gritsman et al., 2000*; *Hagos and Dougan, 2007*). In this model, induction of long-range genes only requires Nodal for short periods of time, whereas activation of short-range genes depends on an extended period of exposure to high Nodal levels. It has therefore been suggested that the total cumulative dose of Nodal signaling determines the cell fate but it is unclear at which level in the pathway a cumulative dose would be measured (*Hagos and Dougan, 2007*).

Studies of TGFβ signaling in other contexts have suggested additional mechanisms for the time-dependent interpretation of Nodal signaling. In *Xenopus*, analysis of signaling by Activin, a TGFβ signal related to Nodal, has suggested a ratchet model: the response to the signal is maintained once the ligand has bound the receptor. Indeed, a short pulse of Activin is sufficient to induce and maintain target gene expression several hours after the pulse (*Gurdon et al., 1995*, *1998*; *Dyson and Gurdon, 1998*; *Bourillot et al., 2002*). This molecular 'memory' has been shown to rely on the persistence of active receptor-ligand complexes (*Jullien and Gurdon, 2005*) and allows changes in signaling output only in response to increasing Activin concentrations but not to decreasing concentrations.

Cell culture studies have suggested that time-dependent Nodal interpretation is dictated by the dynamics of the signaling pathway (*Inman et al., 2002*; *Xu et al., 2002*; *Nicolás et al., 2004*; *Schmierer and Hill, 2005*; *Guzman-Ayala et al., 2009*). TGFβ signaling pathways operate through distinct steps: ligand binding to its receptor, phosphorylation and nuclear accumulation of Smad2, and induction of target gene expression (*ten Dijke and Hill, 2004*; *Massagué, 2012*). Several studies have revealed parameters that affect the levels of activated Smad2. For example, cultured human keratinocytes take approximately 60 min of ligand exposure to generate the maximum level of activated Smad2 (*Inman et al., 2002*). Other studies have shown that the rates of Smad2 phosphorylation and nucleo-cytoplasmic transport affect signaling output (*Clarke et al., 2006*; *Zi and Klipp, 2007*; *Schmierer et al., 2008*; *Vizán et al., 2013*) or that the speed and frequency of TGFβ ligand presentation influences target gene response (*Sorre et al., 2014*). These cell culture studies highlight the potential roles of signaling dynamics in target gene induction but it is unclear how these dynamics affect the response to Nodal in vivo.

To distinguish between the numerous proposed mechanisms for Nodal morphogen interpretation, we studied the temporal and spatial dynamics of Smad2 activation and target gene induction in the early zebrafish embryo. We find that not only Nodal concentration and time of exposure but also the kinetics of target gene induction are key determinants of the response to Nodal morphogens. In particular, our study indicates that a target gene's transcription rate and onset of activation are major determinants of expression range, revealing previously unrecognized layers in the interpretation of morphogen gradients.

## Results

### Smad2 is essential for Nodal signaling

Smad2 activation has been used as a read-out for Nodal signaling, but it has been unclear whether this transcriptional regulator is the main transducer of Nodal signaling in zebrafish (*Dick et al., 2000*; *Jia et al., 2008*). To test the role of Smad2 in Nodal signal transduction, we used TILLING (*Wienholds et al., 2003*) to recover a non-sense mutation in *smad2* and generated embryos lacking maternal and zygotic Smad2 (MZ*smad2*; *Figure 1A,B*). Endoderm and head and trunk mesoderm are absent in MZ*smad2* embryos, a phenotype very similar or identical to Nodal loss-of-function mutants (*Figure 1C–E*) (*Feldman et al., 1998*; *Gritsman et al., 1999*). MZ*smad2* mutants could be rescued by ubiquitous expression of wild-type Smad2 and GFP-Smad2 and the larval lethality of Z*smad2* mutants could be rescued to adulthood by a GFP-Smad2 transgene (*Figure 1F–H*; *Table 1*). Moreover, neither Nodal nor Activin displayed any activity in MZ*smad2* mutants (*Figure 1I,J*).

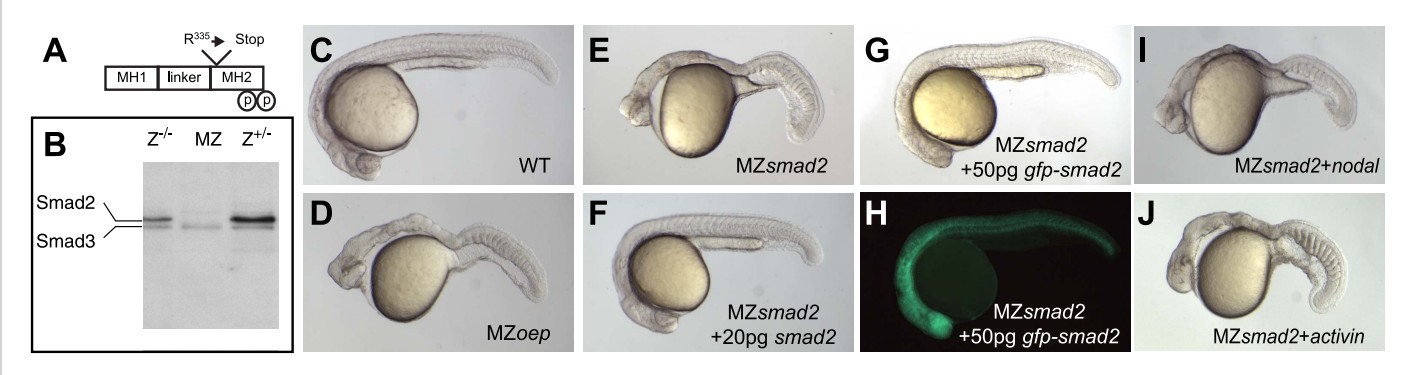

**Figure 1**. Maternal Smad2 is necessary for mesendoderm specification by Nodal signaling. (**A**) Illustration of the Smad2 protein showing the position of the ENU-induced non-sense mutation. (**B**) Western blot against Smad2/3 on 24 hpf embryos of different genotypes for *smad2*. MZ, maternal-zygotic homozygotes, $Z^{-/-}$, zygotic homozygotes, $Z^{+/-}$, zygotic heterozygotes. The pool of maternally contributed Smad2 protein persists for at least 24 hr in zygotic homozygous embryos while it is depleted in MZ*smad2* mutants. (**C–J**) Phenotypic analysis of 36 hpf zebrafish embryos. (**C**) Wild-type embryo. (**D**) MZ*oep* embryo: maternal-zygotic mutant for *one-eyed pinhead* (*oep*), a cell surface protein required for Nodal signaling (*Gritsman et al., 1999*). (**E**) MZ*smad2* embryo. M*smad2* mutants display a very similar phenotype (not shown). (**F**) MZ*smad2* embryo rescued with 20 pg of *smad2* mRNA. (**G–H**) MZ*smad2* embryo rescued with 50 pg of *gfp-smad2* mRNA (brightfield (**G**), epifluorescence (**H**)). *smad2* mRNA appears to be more effective in rescuing the prechordal plate defects in MZ*smad2* mutants as compared to *gfp-smad2* mRNA. (**I**) MZ*smad2* embryo injected with 5 pg mRNA for the zebrafish Nodal homolog *squint*. (**J**) MZ*smad2* embryo injected with 5 pg mRNA for *activin*. Note that while Activin can activate the Nodal pathway in the absence of *oep* (*Gritsman et al., 1999*; *Cheng et al., 2003*), neither Squint nor Activin can activate the pathway in the absence of Smad2.

These results demonstrate that Smad2 is an essential transducer of Nodal signaling during mesendoderm specification.

## Spatio-temporal map of Smad2 activity and target gene expression

The nuclear accumulation of GFP-Smad2 is a well-established reporter of TGFβ signaling (*Nicolás et al., 2004*; *Xu and Massagué, 2004*; *Harvey and Smith, 2009*). This approach has been applied in embryos to visualize how the activated Smad2 gradient evolves over time (*Harvey and Smith, 2009*), but it has not yet been determined how Smad2 activity changes in individual cells and how cell movements might influence gradient interpretation (*Xiong et al., 2013*) (*Figure 2A*). We therefore generated stable transgenic lines in which both GFP-Smad2 and histone H2B-RFP were ubiquitously expressed (*Figure 2B,C*), and tracked GFP-Smad2 nuclear accumulation at the single cell level over time and space.

To enable accurate quantification, we determined how Smad2 phosphorylation, GFP-Smad2 phosphorylation and GFP-Smad2 nucleo-cytoplasmic (NC) ratio increased with increasing Nodal signaling (*Figure 2—figure supplement 1*). These calibrations established that the GFP-Smad2 NC ratio could serve as a read-out for pathway activity and confirmed the graded nuclear accumulation of Smad2-GFP along the vegetal–animal axis (*Harvey and Smith, 2009*) (*Figure 2D*, *Figure 2—figure supplement 1*).

**Table 1**. β-actin::GFP-Smad2 transgene rescues *smad2/smad2* adult lethality

| | smad2/+ X smad2/+; Tg(GFP-Smad2)/+ | |
|---|---|---|
| Genotype | +/+ | Tg(gfp-smad2)/+ |
| +/+ | 11 (37%) | 4 (15%) |
| smad2/+ | 19 (63%) | 17 (60%) |
| smad2/smad2 | 0 (0%) | 7 (25%) |

*smad2/+* fish were crossed to *smad2/+; Tg(GFP-Smad2)/+* fish and their progeny was raised to adulthood and genotyped for *smad2* and for *Tg(GFP-Smad2)*. The only recovered adult progeny homozygous for *smad2* contains a copy of the *GFP-Smad2* transgene.

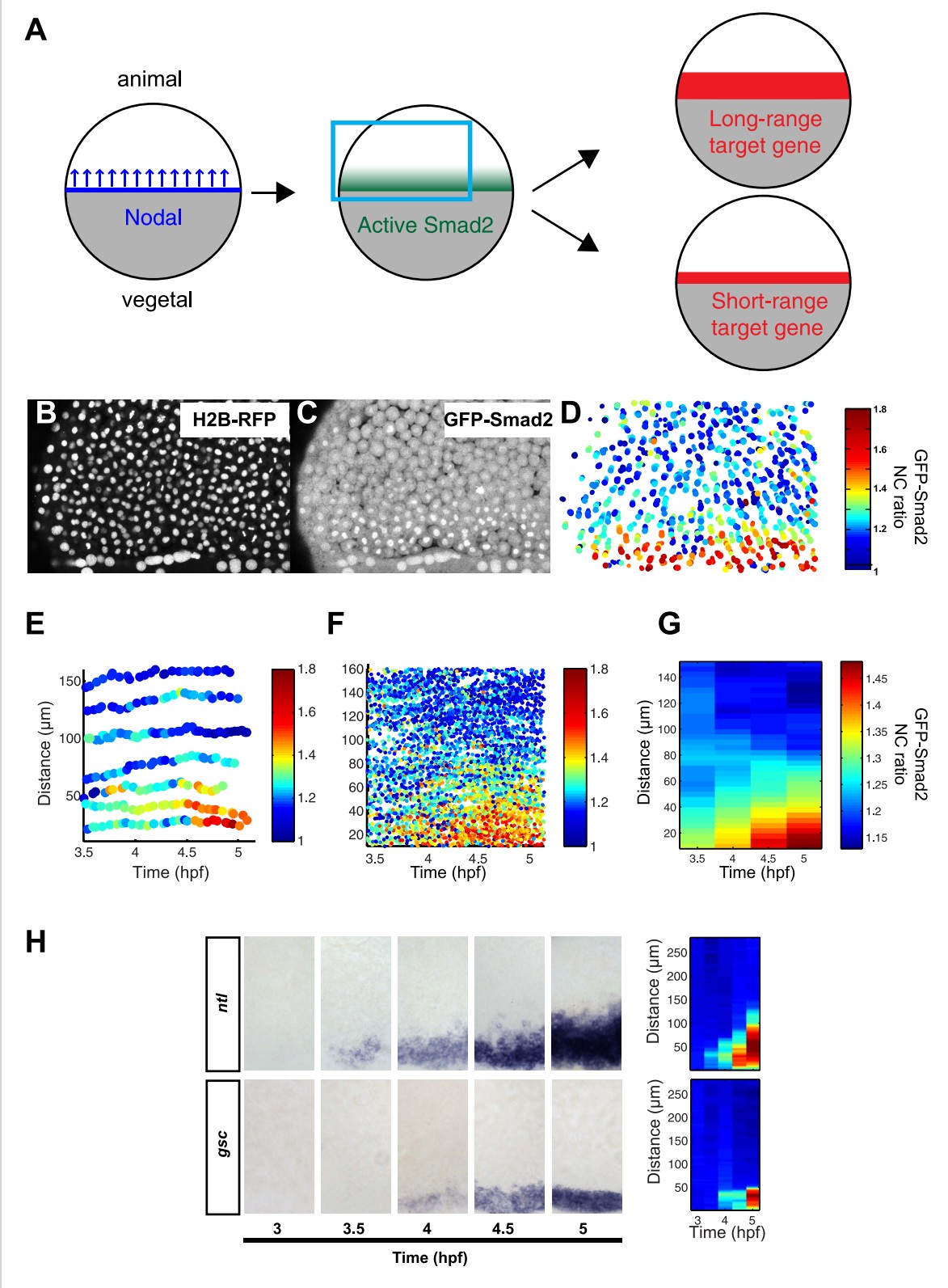

**Figure 2**. Dynamics of Nodal signaling in vivo. (**A**) Illustration of Nodal signaling input–output relationship during blastula stage. Gray = yolk, white = blastoderm. Nodal is produced at the margin, diffuses and forms a gradient along the vegetal–animal axis. Nodal signaling induces a gradient of activated Smad2, which induces long- and short-range target gene expression. (**B** and **C**) Maximal intensity projection of a confocal stack of a Histone 2B-RFP (**B**), GFP-Smad2 (**C**) double transgenic embryo at 50% epiboly (blue box in (**A**)). GFP-Smad2 strongly accumulates in the nuclei of cells close to the

*Figure 2. continued on next page*

*Figure 2. Continued*

margin, the source of Nodal signals. (**D**) Heatmap of the nucleo-cytoplasmic (NC) ratio of GFP intensity from the embryo in (**B** and **C**). Each dot represents the position of a cell (overlay of five consecutive frames, 3-min intervals per frame). Each cell is color-coded according to its GFP NC ratio (see *Figure 2—figure supplement 2* for movement of cells). (**E**) Examples of single cell tracks at different locations along the vegetal–animal axis, showing changes in GFP-Smad2 NC ratio over time. The position of most cells relative to the margin remains constant during blastula stage. Cells close to the margin activate Nodal signaling earlier and at higher levels than cells at a distance from the margin. The short bursts observed in some cell tracks are caused by transient nuclear accumulation of GFP-Smad2 at the onset of nuclear envelope breakdown and are observed even in the absence of Nodal signaling. (**F**) NC ratio dynamics of tracked cells along the vegetal–animal axis. (**G**) Mean NC ratio values from (**F**) in 30 min bins. Note that the range and amplitude of the Smad2 activity gradient increase over the course of 90 min. Basal NC ratio is higher in younger embryos (see *Figure 2G*, 3.5 hpf). Since this phenomenon is also observed in the absence of Nodal signaling (MZ*oep* mutants), the higher NC ratio is unlikely to reflect early Smad2 activation, but a higher nuclear import/export ratio of GFP-Smad2 during early development. (**H**) Time course of *ntl* (upper panel) and *gsc* (bottom panel) expression detected by RNA in situ hybridization. *ntl* begins to be induced as early as 3.5 hpf and its domain of expression expands over time to 100–120 μm from the margin; *gsc* begins to be induced 30 min later than *ntl* and its domain of expression expands to 50 μm from the margin. Close-up views of dorsal side, animal pole to the top. Right panel, heatmap for the grayscale intensity of in situ hybridization signals along the vegetal–animal axis showing the increase in range and intensity of *ntl* and *gsc* expression over time. See *Figure 2—figure supplement 3* for comparison of probes and *Figure 2—figure supplement 4* for independent validation of *gsc* and *ntl* expression domains using Seurat.

The following source data and figure supplements are available for figure 2:

**Source data 1**. Individual cell tracks and NC ratio.

**Figure supplement 1**. GFP-Smad2 as a sensor of Nodal activity in vivo.

**Figure supplement 2**. Cell movements during blastula stages.

**Figure supplement 3**. Detection sensitivity of *ntl* and *gsc* by in situ hybridization.

**Figure supplement 4**. Comparison of *ntl* and *gsc* expression pattern from single-cell RNAseq analysis.

To follow the trajectory of each cell, we tracked individual blastomeres over time (*Figure 2D–F*, *Figure 2—figure supplement 2*, *Figure 2—source data 1*), determined their GFP-Smad2 NC ratio, and measured their distance from the margin. The resulting spatio-temporal map of Smad2 activity revealed that (1) the position of cells relative to the margin did not change extensively until the onset of gastrulation (*Figure 2E*); (2) cells close to the margin tended to activate Smad2 early and reached the highest levels of activated Smad2; (3) cells located farther away from the margin tended to activate Smad2 with a delay and the levels of activated Smad2 remained low (*Figure 2F,G*). Thus, during the 1.5 hr from mid- to late-blastula stage a low-amplitude short-range gradient of activated Smad2 is transformed into a high-amplitude long-range gradient.

To determine how the expression range of Nodal target genes correlates with Smad2 activity, we analyzed the expression of the long-range and short-range genes *ntl* and *gsc*, respectively (*Figure 2H*, *Figure 2—figure supplements 3, 4*). *ntl* was first faintly detected in a few cells on the presumptive dorsal side of the embryo at the mid-blastula stage. Subsequently, its expression domain intensified and progressively extended animally until the onset of gastrulation. By contrast, *gsc* expression initiated ~30 min later and remained confined to a narrow domain on the dorsal side (*Figure 2H*). Comparing the spatio-temporal maps of Nodal target gene expression and Smad2 activity confirmed that the long-range target gene *ntl* was induced at both high and low levels of activated Smad2, whereas the expression of the short-range gene *gsc* correlated with high Smad2 levels and sustained Smad2 activity.

## Testing the threshold and ratchet models

The spatiotemporal maps of Smad2 activity and target gene expression are consistent with previous proposals postulating that signaling thresholds determine target gene induction (*Harvey and Smith, 2009*). To directly test the threshold model of Nodal signaling, we wished to determine whether high Smad2 activity fully predicts the activation of both short- and long-range Nodal target genes. Using transplantation assays, we exposed GFP-Smad2 cells to high Nodal levels for different periods of time and analyzed the relationships between activated Smad2 levels and target gene expression (*Figure 3A*).

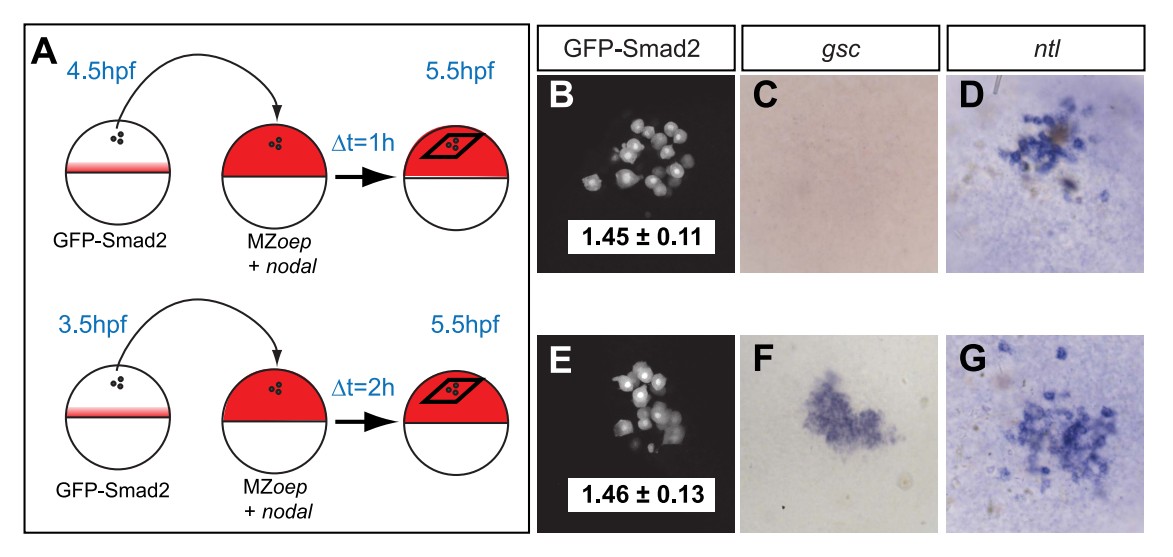

**Figure 3**. Testing the threshold model. (**A**) Schematic of the transplantation experiment. Animal pole cells (black circles) from a GFP-Smad2 transgenic embryo were transplanted into the animal pole of a host embryo that had been injected with mRNA for *squint*, a zebrafish Nodal gene (red). Host cells were unresponsive to Nodal because they were maternal-zygotic mutants for *one-eyed pinhead* (MZ*oep*), a cell surface protein required for Nodal signaling. This strategy prevents feedback loops and restricts target gene expression to donor cells. The developmental age of donor cells was matched to host embryos. Black parallelograms indicate imaging plane in subsequent panels. (**B–G**) Nodal signaling response of donor cells after 1 hr (**B–D**) or 2 hr (**E–G**) of exposure to Nodal. (**B** and **E**) Projection of confocal stacks of transplanted embryos and associated NC ratio (mean ± std). Activated Smad2 levels are similar in both cases. See *Figure 3—figure supplement 1* for time course of GFP-Smad2 N/C ratio. (**C** and **F**) RNA in situ hybridization for *gsc*. (**D** and **G**) RNA in situ hybridization for *ntl*. *ntl* is expressed after 1 (n = 12/12) or 2 hr (n = 16/16) of Nodal exposure while *gsc* signal in transplanted cells is only detected after 2 hr of exposure (n = 1/15 at 1 hr, n = 12/14 after 2 hr). Images in **B–G** are from different embryos. Note that the differences in the duration of Nodal exposure uncouple the activated Smad2 level from target gene expression.

The following figure supplements are available for figure 3:

**Figure supplement 1**. Time course of GFP-Smad2 NC ratio.

**Figure supplement 2**. Testing the ratchet model.

GFP-Smad2 NC ratios were similar in cells exposed to Nodal for either 1 or 2 hr (*Figure 3B,E*). However, while the long-range gene *ntl* was expressed both after one or 2 hr of exposure to Nodal (*Figure 3D,G*), the expression of the short-range gene *gsc* was only detected after 2 hr of exposure (*Figure 3C,F*). These results are inconsistent with the strictest forms of the threshold model—the level of Smad2 activity at a given time predicts target gene expression—and reveal that the duration of signaling influences morphogen interpretation primarily at the level of target gene induction.

The spatiotemporal maps of Smad2 activity and target gene expression support one prediction of the ratchet model—cells respond to increases in ligand concentration. To test the other tenet of the ratchet model—cells remember the highest ligand concentration they have been exposed to—we determined whether response is refractory to decreasing Nodal levels. We transplanted GFP-Smad2 cells from the blastula margin (where Nodal concentration is high) to the animal pole (where Nodal concentration is low). Inconsistent with the ratchet model, Smad2 activity progressively decreased and reached basal levels after ~60 min (*Figure 3—figure supplement 2A*). Similarly, the expression of the long-range gene *ntl* disappeared over time (*Figure 3—figure supplement 2B*). Thus, pathway activity and target gene expression cannot be maintained for extended periods after transient exposure to Nodal.

## A kinetic model for Nodal morphogen gradient interpretation

Since the threshold and ratchet models do not fully account for Nodal morphogen interpretation, we sought an alternative model based on the biochemistry and biophysics of signaling. The changes in Smad2 activity and gene expression suggested that the kinetics of signal transduction and gene

induction might be major factors in Nodal morphogen interpretation. To determine how time and concentration might translate into pathway activity and target gene response, we developed a mathematical description of the kinetics of Nodal signaling (*Chen et al., 2010*) (*Source code 1*). To reduce the complexity of the system and the numbers of free parameters, we focused on the three main steps in the pathway (*Figure 4A*): (1) the diffusion of Nodal from a local source, (2) the Nodal-dependent phosphorylation of Smad2 (pSmad2), and (3) the pSmad2-dependent transcription of target genes. Three coupled differential equations were formulated to implement the kinetic model. All equations were based on standard reaction-diffusion models and mass-action kinetics.

$$\frac{\partial N}{\partial t} = P(x,t) + D_N.\nabla^2.N - k_1.N. \tag{1}$$

*Equation 1* describes the change of Nodal (N) levels over time. Nodal is produced from a source, diffuses and is degraded. Nodal levels at a distance from the source increase with increases in Nodal production (P) and diffusion (D) and with decreases in clearance ($k_1$).

$$\frac{dS_p}{dt} = k_2.N.S - k_3.S_p. \tag{2}$$

*Equation 2* describes the change in activated (phosphorylated) Smad2 ($S_p$) levels over time. Smad2 activation is proportional to Nodal and non-activated (non-phosphorylated) Smad2 (S) concentrations. Thus, when Nodal concentration increases, activated Smad2 levels increase. Smad2 is deactivated (de-phosphorylated) at rate $k_3$.

$$\frac{dRNA_{target}}{dt} = \alpha.\frac{S_p^n}{K_d^n + S_p^n} - \beta.RNA_{target}. \tag{3}$$

*Equation 3* describes the induction of Nodal target genes ($RNA_{target}$) over time. For each target gene, levels of expression and dynamics of induction are defined by its maximal transcription rate ($\alpha$), degradation rate of its RNA ($\beta$), and the affinity of pSmad2 for its promoter/enhancer ($K_d$). The expression of a given target gene increases as $\alpha$ increases, $K_d$ decreases, or the degradation rate decreases. As the concentration of pSmad2 increases, target gene transcription increases. The Hill coefficient $n$ defines the cooperativity that modulates the sensitivity of the response.

## Constraining the kinetic model through in vivo measurements

To test the effectiveness of the kinetic model in explaining and predicting Nodal gradient interpretation, we wished to run simulations with a realistic set of parameters. The effective diffusion coefficients and clearance rates of Nodals have been experimentally determined (*Müller et al., 2012*), but other parameters of the system have not been measured. Exploring the contribution of each of these parameters in regulating target gene expression revealed that multiple parameter combinations could simulate the expression patterns observed in vivo (*Figure 4B* and *Figure 4—figure supplement 1*). In particular, the range of expression is affected most dramatically by changes in transcription rate, $K_d$ or Hill coefficient. We therefore decided to constrain the parameter space by performing a detailed quantification of pSmad2 levels and target gene expression at different Nodal concentrations and durations of exposure. To precisely control the levels and timing of ligand input, we injected different amounts of recombinant mouse Nodal protein into the extracellular space of blastula embryos that lacked endogenous Nodal ligands (*Figure 5A*). Nodal-injected embryos were collected at different time points and pSmad2 levels were determined by Western blotting. Target gene expression levels were measured by NanoString analysis using a custom-designed codeset (*Figure 5—source data 1*). This technique combines fluorescently barcoded probes with microimaging to detect and count hundreds of transcripts simultaneously in a single hybridization reaction and without amplification. It thus avoids the primer-specific amplification biases of qRT-PCR experiments and allows the direct measurement and comparison of transcript levels (*Geiss et al., 2008*; *Su et al., 2009*; *Strobl-Mazzulla et al., 2010*).

### Phospho-Smad2

As predicted by the kinetic model and in agreement with previous studies (*Zi et al., 2011*), increasing Nodal levels induced higher levels of pSmad2 until a plateau was reached. High Nodal levels lead to

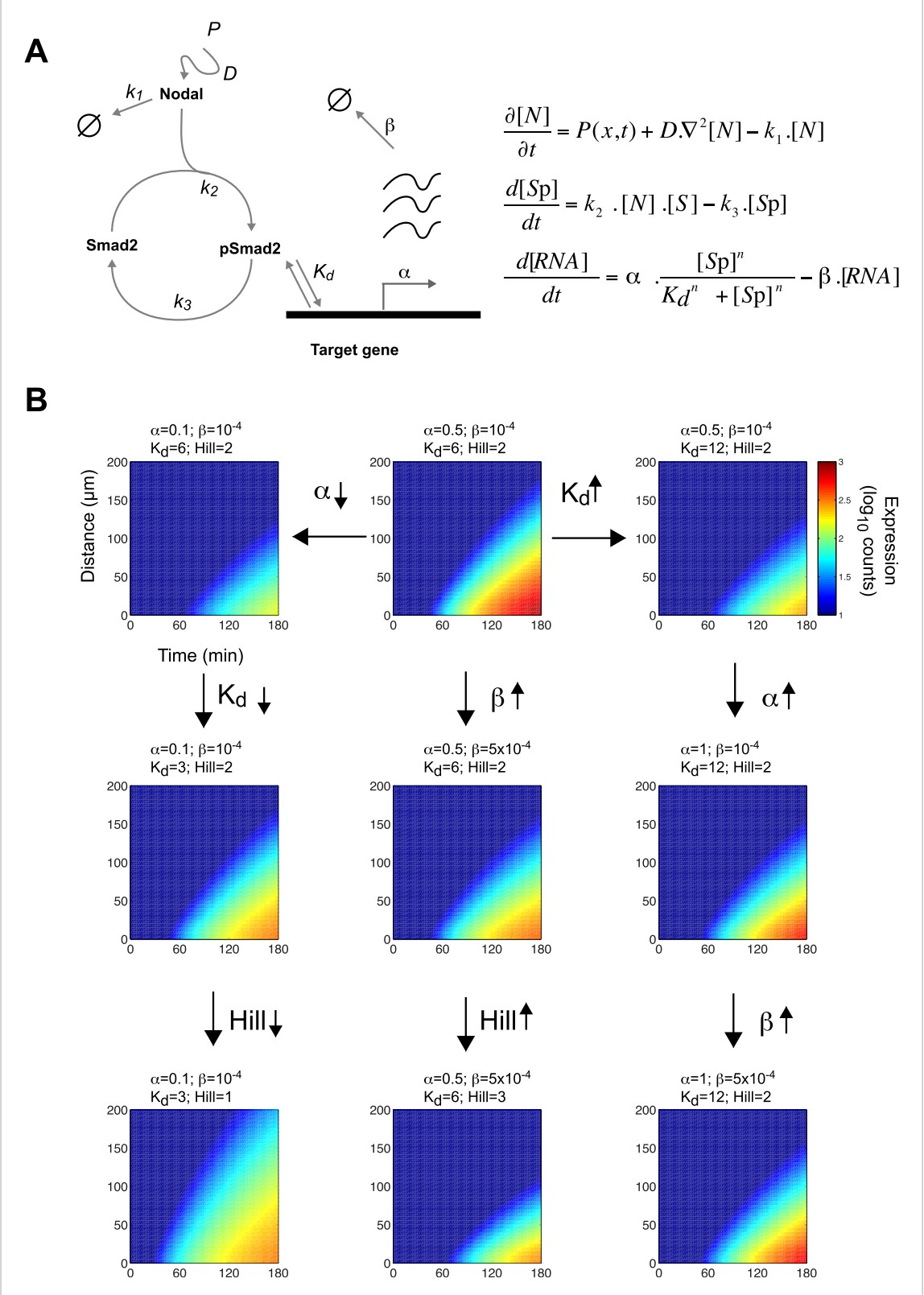

**Figure 4**. A kinetic model for Nodal morphogen interpretation. (**A**) Diagram of the Nodal signaling pathway used for modeling (left) and coupled differential equations describing the changes of Nodal, activated Smad2 and target genes over time (right). The Nodal ligand is locally produced, diffuses and via kinase receptors phosphorylates Smad2. Phosphorylated Smad2 acts as a transcription regulator and binds to target genes to induce

*Figure 4. continued on next page*

*Figure 4. Continued*

transcription. (**B**) Spatiotemporal gene expression patterns were simulated over 3 hr using the kinetic model. Each panel depicts the expression pattern resulting from a unique combination of the four free parameters involved in mRNA production (transcription rate α, degradation rate β, dissociation constant $K_d$ and Hill coefficient) while other parameters are held constant. Note how changes in these parameters change the range of target gene expression. See *Figure 4—figure supplement 1* for more extensive simulations.

The following figure supplement is available for figure 4:

**Figure supplement 1**. Screening for parameters regulating range and onset of target gene expression.

a very rapid activation of Smad2, reaching a quasi-steady state after 1 hr. In contrast, low Nodal levels induced lower levels of Smad2 activation and quasi-steady state Smad2 activation was only reached 2 hr after ligand exposure (*Figure 5B*).

We subjected the kinetic model to a fitting procedure to identify values that would best reflect the experimental data (see 'Materials and methods'). For Smad2 activation, we found phosphorylation rates (range from $2.3 \times 10^{-6}$ to $4.0 \times 10^{-6}$ nM$^{-1}$s$^{-1}$) and turnover rates (range from $0.9 \times 10^{-4}$ to $2.7 \times 10^{-4}$ s$^{-1}$) similar to previous studies performed in cell culture and *Xenopus* animal cap cells (*Bourillot et al., 2002*; *Schmierer and Hill, 2005*; *Schmierer et al., 2008*).

## Target gene expression

To measure target gene expression, we first identified genes in our NanoString codeset that were directly regulated by Nodal signaling using three criteria: (1) increased expression upon increase of Nodal levels (*Figure 5C*), (2) Nodal-mediated gene induction in the presence of translation-blocking cycloheximide (*Figure 5—figure supplement 1*), and (3) binding by Smad2 in the vicinity of transcription start sites (often in conjunction with the co-regulator FoxH1, *Figure 5—figure supplement 1*, *Figure 5—source data 2–5*) (*Liu et al., 2011*; *Yoon et al., 2011*). This analysis identified 47 direct targets of Nodal signaling.

NanoString analysis allowed precise comparisons of transcript levels in response to different levels and duration of Nodal exposure (*Geiss et al., 2008*; *Strobl-Mazzulla et al., 2010*; *Nam and Davidson, 2012*). Target genes had specific response profiles (*Figure 5C*, *Figure 5—source data 6*). For example, *ntl*, a typical long-range target, was induced at low Nodal concentrations and its expression reached high NanoString counts (~2500) at high Nodal concentrations. The induction of *ntl* expression was rapid: *ntl* mRNA accumulated within 30 min after injection of intermediate levels of Nodal and continuously increased over time. By contrast, *gsc*, a short-range Nodal target, required higher concentrations of Nodal to be detected above background levels, and its NanoString counts at high Nodal concentrations were 25 times lower than those of *ntl*. The induction of *gsc* was slow: mRNA accumulation was only detected after 60 min. These measurements reveal striking differences in the transcriptional magnitude and timing of Nodal-induced gene expression.

To examine whether the kinetic model could capture the behavior of individual target genes, we screened for gene-specific parameter combinations that satisfied the constraints imposed by the NanoString measurements (see 'Materials and methods', *Source code 2*). Parameter value search was limited to defined intervals: $10^{-3}$ to $10^1$ counts per second for the transcription rate (*Miller et al., 2011*; *Tu et al., 2014*), 0.1 to 100 nM for $K_d$ (*Xi et al., 2011*; *Geertz et al., 2012*; *Gentsch et al., 2013*), $1 \times 10^{-5}$ to $1 \times 10^{-3}$ s$^{-1}$ for the degradation rate (*Rabani et al., 2011*) and Hill coefficients from 1 to 4. Reflecting the different transcript levels measured by NanoString, transcription rates varied widely between genes, ranging from 0.0016 to 9.3 counts/s (mean 0.54 ± 1.66). In contrast, $K_d$s only ranged from 0.73 to 42 nM, with more than 75% of $K_d$s between 5 and 10 nM (*Figure 5—source data 7*). For example, we found $K_d$s of Smad2 for *ntl* and *gsc* of 4.9 nM and 5.4 nM, respectively, while the transcription rates were 0.67 counts/s for *ntl* and 0.032 counts/s for *gsc*. These differences explain why only prolonged exposure to Nodal induced *gsc* in the test of threshold model (*Figure 3*): only very low *gsc* RNA counts (~50) can be detected 1 hr after Nodal exposure. In contrast, *ntl*, which was rapidly induced in the test of the threshold model, was induced at high levels (~1000 counts) after 1 hr. The finding that a realistic set of parameter combinations satisfied the constraints imposed by the NanoString measurements suggests that the kinetic model provides a suitable description of Nodal morphogen interpretation.

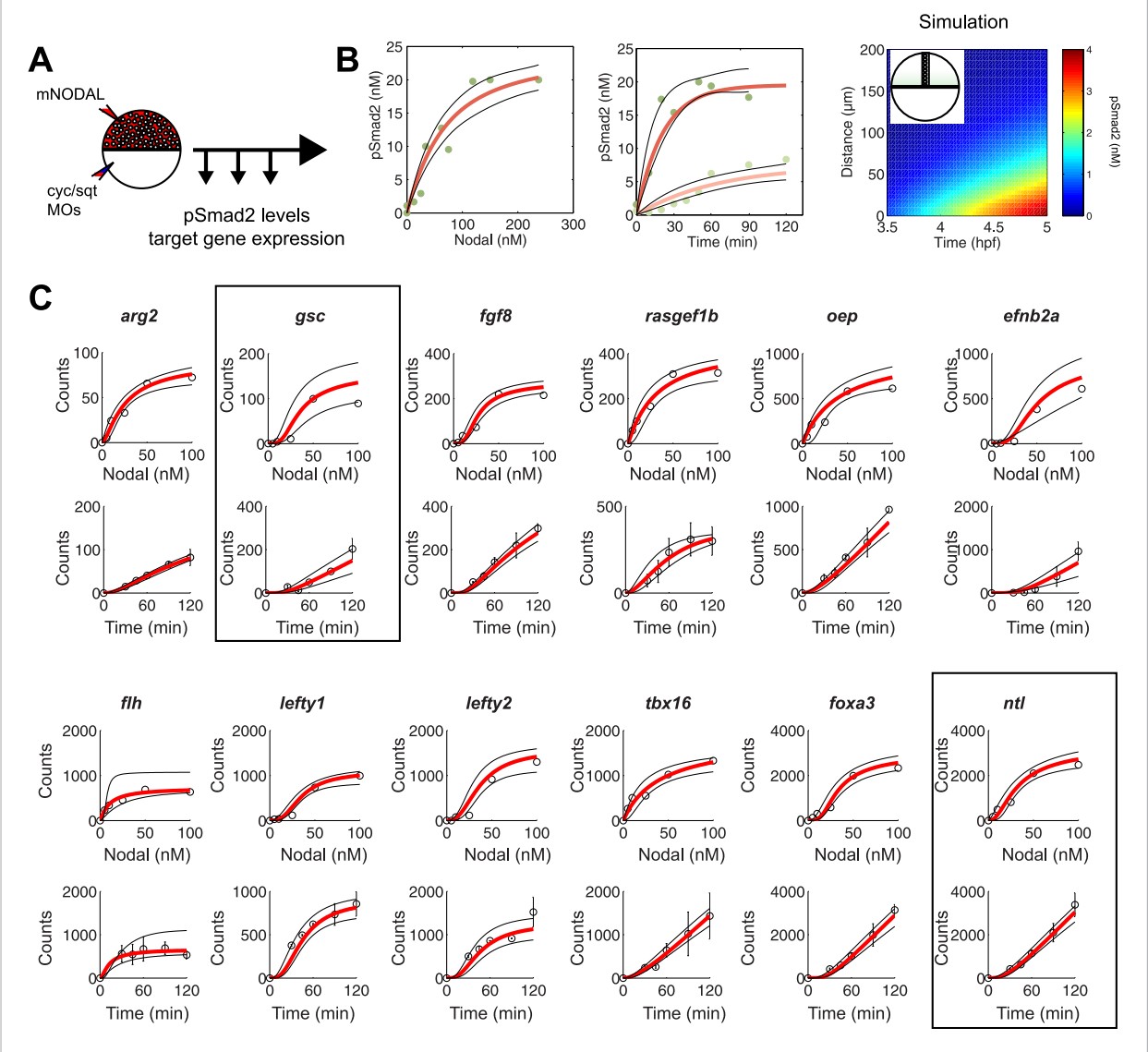

**Figure 5**. Constraining the kinetic model through in vivo measurements. (**A**) Experimental design: Wild-type embryos were injected at the one-cell stage with *squint* and *cyclops* MOs to knock down endogenous Nodal signaling. Morphant embryos were further injected either at 3.5 or 4.5 hpf with recombinant mouse Nodal protein at different concentrations in the extracelluar space. They were then incubated for different periods of time and processed for Western blot to determine pSmad2 levels or for NanoString to assess mRNA levels. (**B**) Dose-response (left panel) and time course (middle panel) of Smad2 activation at high (100 nM, dark green) and low (10 nM, light green) Nodal concentrations. Dots represent experimental data points and orange lines show model simulations with $k_2 = 3.13 \times 10^{-6}$ nM$^{-1}$s$^{-1}$ and $k_3 = 1.8 \times 10^{-4}$ s$^{-1}$. Black lines represent the 95% confidence intervals of data predictions. (Right panel) Simulated spatial distribution of Smad2 activation in a one-dimensional column of cells from 3.5 to 5 hpf in response to Nodal production from a source that extends from L = 0 to 25 µm. (**C**) Dose-response (top) and time course (bottom) data of 12 direct Nodal targets (black dots). Given a specific set of parameters for each gene, the model (red line) recapitulates the dynamics of gene expression. Black lines represent fits encompassing the 95% prediction confidence intervals. *gsc* and *ntl* dynamics are highlighted within black boxes.

The following source data and figure supplements are available for figure 5:

**Source data 1**. NanoString Probeset.

**Source data 2**. Smad2 associated peaks after Nodal injection.

**Source data 3**. Smad2 associated peaks after Nodal signaling inhibition.

**Source data 4**. FoxH1 associated peaks after Nodal injection.

*Figure 5. Continued*

**Source data 5**. FoxH1 associated peaks after Nodal signaling inhibition.

**Source data 6**. NanoString counts of Nodal target genes.

**Source data 7**. Nodal target genes identified in the NanoString codeset and their associated characteristics.

**Figure supplement 1**. Characterization of direct Nodal target genes.

## The kinetic model predicts the range of target gene expression

To test whether the kinetic model can recapitulate and predict the temporal and spatial pattern of Nodal target gene expression, we ran simulations in a one-dimensional column of cells spanning the vegetal–animal axis. We let Nodal be produced and diffuse from a point source and used the parameters identified in the previous section to simulate the spatial Smad2 activation and transcriptional response over time. Using these conditions, the spatiotemporal pattern of activated Smad2 correlated well with the endogenous pattern of Smad2 activation (*Figure 5B*): a short-range low-amplitude gradient was transformed over time into a long-range high-amplitude gradient, as observed in vivo for the GFP-Smad2 activity gradient (*Figure 2F,G*) (*Harvey and Smith, 2009*).

The simulated spatiotemporal patterns of gene expression also fit well with the in vivo data. For example, in our simulations, *ntl* expression began in cells close to the margin 45 min after ligand production started, and the range of *ntl* continuously increased and reached cells located more than 100 μm away from the margin after 3 hr (*Figures 2H, 6A*). By contrast, *gsc* expression was delayed and its range of expression was confined to cells close to the source (*Figures 2H, 6B*).

To test the predictive power of the kinetic model, we determined the expression patterns of genes that had not been analyzed in detail with respect to their range. As predicted by the simulations, *foxa3* mRNA rapidly accumulated at high levels up to four cell tiers (~60 μm) from the source and then extended up to 80–100 μm by the onset of gastrulation (*Figure 6C*). *efnb2a* mRNA also readily accumulated but was expressed in a narrower domain, as predicted from the simulations (*Figure 6D*). These results reveal the power of the kinetic model in recapitulating and predicting the response of target genes to Nodal morphogen signaling.

## Transcription rate predicts range of target gene expression

Since the kinetic model predicted target gene expression, we wished to determine which parameters were the major contributors to the range of gene expression. In the simulations described above (*Figure 4B* and *Figure 4—figure supplement 1*), genes whose $K_d$ is low and maximal transcription rate is high are expressed at high levels and at long range. In contrast, the degradation rate influences the range of expression only when mRNA half-lives are very short (*Figure 4—figure supplement 1*). In agreement with the simulations, we found that genes that are highly induced by Nodal generally display a long range of expression (*Figure 7A,B*). Strikingly, the maximal transcription rate, not the $K_d$ or the degradation rate, was the best predictor of gene expression range (*Figure 7C,D*). For example, while the degradation rate, $K_d$ and Hill coefficient for the long-range gene *foxa3* and the short-range gene *gsc* are very similar (*Figure 5—source data 7*), their maximal transcription rate, and therefore their maximal level of expression, differ by a factor of 20. These results raise the possibility that the maximal transcription rate is a major contributor to target gene expression range: the higher the maximal rate of transcription, the longer the range. Moreover, multiple hypotheses analysis indicates that a model in which the maximal transcription rate is gene-specific and $K_d$ is identical for all the genes performs better than a model where $K_d$ is gene-specific and the maximal transcription rate is constant (see Nodal signaling modeling section in 'Materials and methods'). Although the $K_d$ may affect target gene response, these analyses indicate that the maximal transcription rate is a key parameter in determining the range of expression.

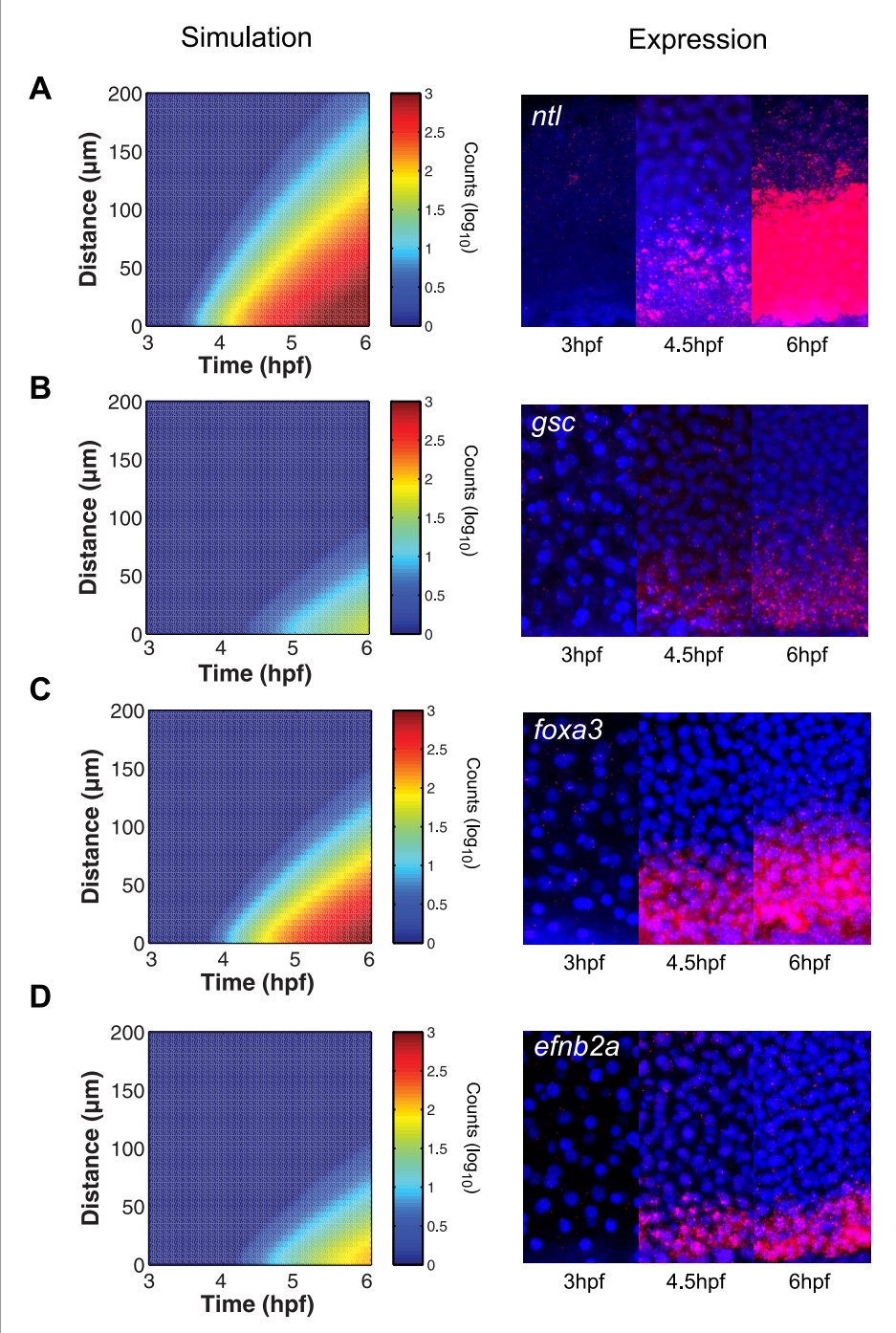

**Figure 6**. The kinetic model predicts gene expression patterns. Comparison of kinetic model simulations and RNA fluorescent in situ hybridization for *ntl* (**A**), *gsc* (**B**), *foxa3* (**C**), *efnb2a* (**D**). Left panels: simulations of spatiotemporal expression patterns over 3 hr along a 200 μm-high column of cells using gene-specific parameters identified in the parameter screen. Right panels: RNA fluorescent in situ hybridization at 3, 4.5 and 6 hpf. The size of the embryonic field is 100 μm wide and 200 μm high. Animal pole to the top.

## Delayed response to Nodal restricts the range of target gene expression

To analyze additional predictions generated by the kinetic model, we asked whether a delay in gene induction might affect target gene response. To simulate this scenario, we extended the kinetic model with a co-factor that is produced later than and independently of Nodal and acts together with Smad2

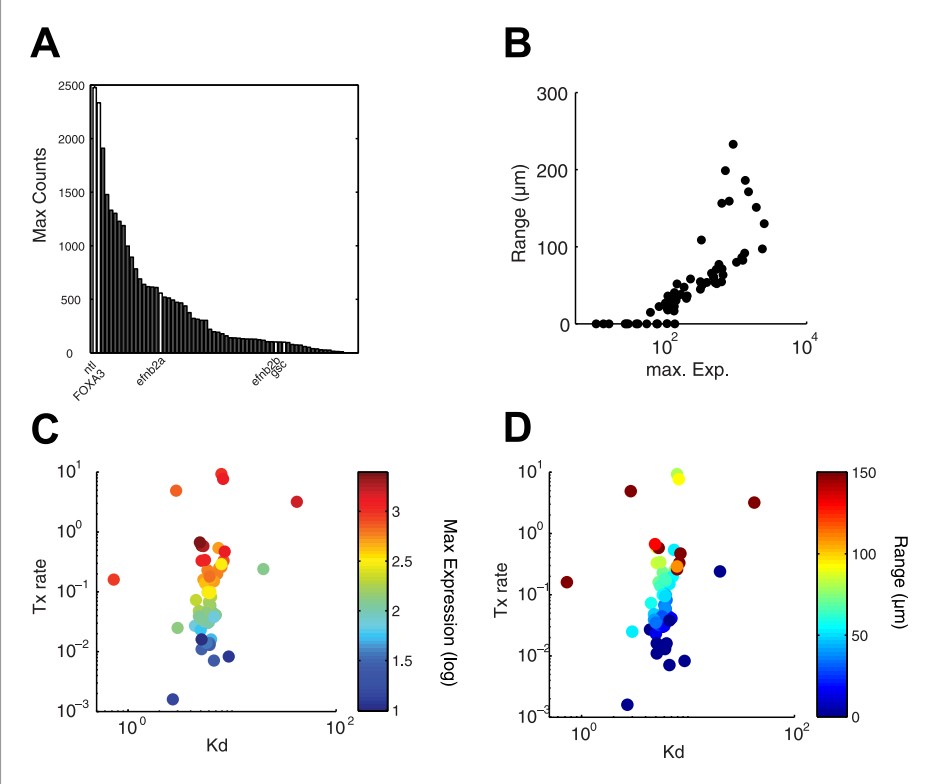

**Figure 7.** Range of expression correlates with maximal transcription rate. (**A**) Bar graph showing the number of counts detected 90 min after injection of 100 nM of recombinant Nodal protein for the 61 Nodal-responsive genes (direct and indirect) identified in the NanoString codeset. Some of the genes used in this study are highlighted. (**B**) Scatter plot comparing maximal expression and simulated range. Highly expressed genes tend to have a longer range of expression. (**C** and **D**) Scatter plots comparing fitted $K_d$ and maximal transcription rate (Tx rate) in relation to maximal expression (**C**) and in relation to simulated spatial range of expression (**D**). Most $K_d$ values remain in a narrow range while transcription rates spread over several orders of magnitude.

to activate gene transcription. Simulations revealed not only the expected delay but also a reduced range of target gene induction: a long-range gene could be transformed into a short-range gene by introducing a delay in gene induction (*Figure 8A*).

To determine whether such delayed genes might exist in vivo, we screened our NanoString data for Nodal targets whose induction upon Nodal exposure was delayed (*Figure 8B*, *Figure 8—figure supplement 1*). We discovered a small set of genes that were induced slowly after Nodal exposure at 3.5 hpf but more rapidly after exposure at 4.5 hpf (*Figure 8B*, *Figure 8—figure supplement 1*). For example, when Nodal was injected at 3.5 hpf, *efnb2b* was only induced after approximately 2 hr. By contrast, when Nodal was injected at 4.5 hpf, the delay in *efnb2b* induction was reduced by more than 30 min (*Figure 8B*). In contrast, most other genes responded rapidly to Nodal exposure at either time point (*Figure 5—figure supplement 1*). This result revealed that the delay was gene-specific and did not reflect a general lack of competence to respond to Nodal signaling or activate gene expression. Similar to the canonical target genes, genes with delayed induction contained Smad2/FoxH1 binding sites and showed a clear response upon injection of Nodal (*Figure 8—figure supplement 1*) but their induction was abolished in the presence of cycloheximide (*Figure 8—figure supplement 1*). These Nodal target genes are therefore likely to be regulated not only by Nodal signaling but additional factors. Strikingly, and as predicted by the delayed induction model, *efnb2b* expression in the embryo was detected only late (6 hpf) and at a short range (5 cell tiers) (*Figure 8C*). Similarly, the Nodal target gene *bra* (*Martin and Kimelman, 2008*) could only be induced shortly before gastrulation and, as predicted by the model, was expressed at low levels and at a short range (*Figure 8—figure supplement 2*). These results reveal that a delay in transcriptional response can be used to limit the range of morphogen-induced gene expression.

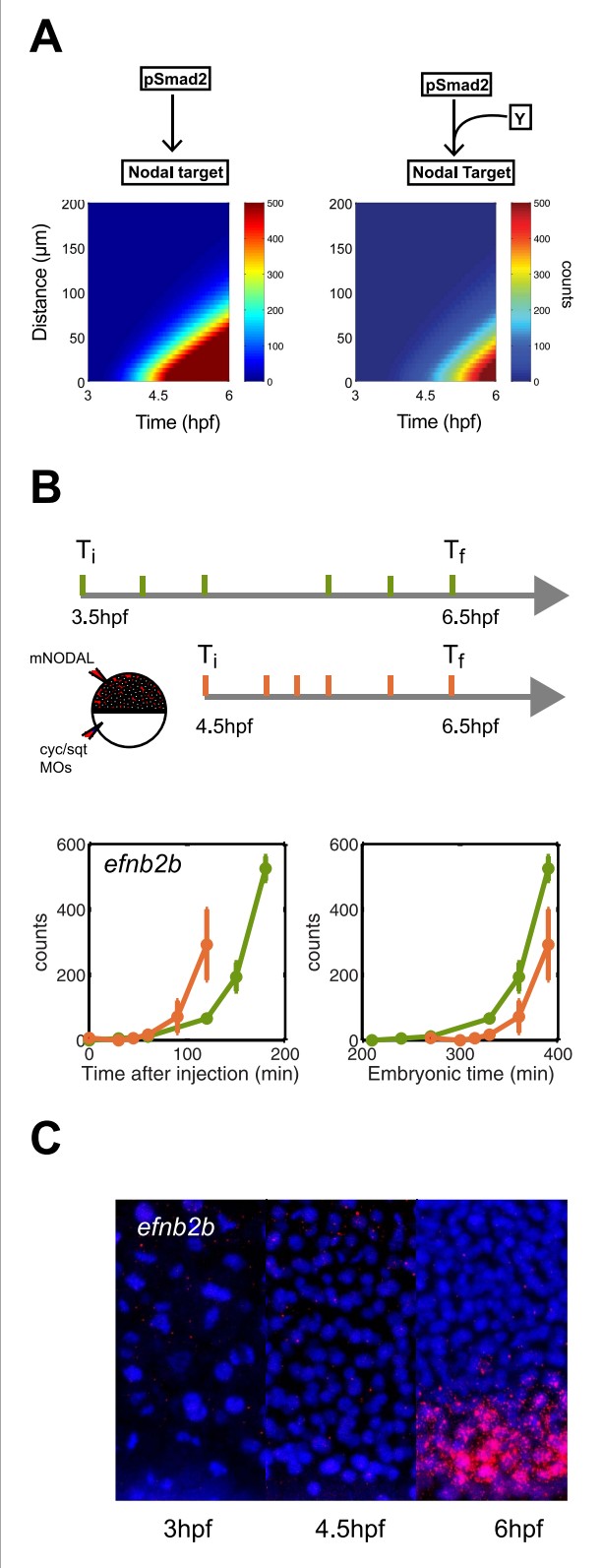

**Figure 8**. Delayed onset of transcription restricts expression range. (**A**) Simulation of *efnb2b* expression using the kinetic model without (left) or with (right) a co-transcriptional activator Y. The dependence on Y delays the onset of *efnb2b* expression and reduces its range.
*Figure 8. continued on next page*

## Discussion

Numerous models have been proposed to explain how morphogen gradients are interpreted to generate diverse gene expression patterns. To interrogate these models, we have taken a quantitative approach to measure the parameters that underlie gradient formation (*Müller et al., 2012*) and interpretation (this study). This approach reveals that the kinetics of target gene induction is a major determinant of morphogen interpretation and suggests that a kinetic model of morphogen interpretation is better suited for the Nodal morphogen system than the prominent threshold and ratchet models.

The kinetic model recapitulates the dynamics of Smad2 activation and reveals how distinct gene expression patterns can be generated: (1) the Nodal morphogen gradient forms and extends through diffusion; (2) rapid phosphorylation generates a corresponding gradient of activated Smad2; target genes are induced based on (3) their affinity for activated Smad2, (4) their maximal transcription rate, and (5) their competence to respond to activated Smad2. Thus, a target gene can be induced rapidly and at a long range by high transcription rate, high Smad2 affinity and early onset of induction. Conversely, low affinity for Smad2, low transcription rate or late onset of induction generate short-range gene expression patterns.

Our analysis identifies transcription rates and induction delays as two novel strategies to modulate morphogen interpretation. Previous models of morphogen interpretation have emphasized the importance of differential DNA (or chromatin) affinity: the higher the affinity for the transcription regulator, the longer the range of target gene expression. Our results do not contradict such models but reveal that in a rapidly developing system, the intrinsic rate of transcription of a target gene can be a major determinant of gene expression range: high affinity binding sites cannot overcome the limits imposed on gene expression range by low levels of

*Figure 8. Continued*

(**B**) Top: Experimental design. Bottom: Time-course induction of *efnb2b* after injecting recombinant Nodal protein at 3.5 hpf (green) and 4.5 hpf (orange). The induction kinetics of this gene are very slow, but the later Nodal is injected, the faster its induction. Note that counts for the expression of late target genes are higher after early injection compared to later injections. This effect might be due to the fact that after early injections phospho-Smad2 levels are high for a longer period before a gene becomes competent to respond as compared to late injections, when there is a shorter time window of high phospho-Smad2 levels. There might be a priming mechanism in which longer exposure to activated Smad2 increases gene expression when competence is reached. (**C**) RNA fluorescent in situ hybridization for *efnb2b* at 3, 4.5 and 6 hpf. Expression of *efnb2b* is only detected at 6 hpf, although Nodal signaling and the expression of most other Nodal targets commences much earlier.

The following figure supplements are available for figure 8:

**Figure supplement 1**. Characterization of co-regulated Nodal target genes.

**Figure supplement 2**. Transcriptional competence regulates the onset and range of *bra* expression.

intrinsic transcription. Similarly, delays in transcriptional onset can turn high affinity target genes into short-range genes.

The Nodal morphogen system stands in contrast to two other well-studied morphogen systems, Sonic Hedgehog (Shh) and Bicoid. The Shh gradient patterns the dorsoventral axis of the mammalian neural tube over several days (*Briscoe and Ericson, 1999*; *Nishi et al., 2009*; *Cohen et al., 2013*). Although different concentrations of Shh elicit different transcriptional responses, feedback and cross-regulatory interactions are key determinants of patterning. For example, cells are progressively desensitized to Shh activity by upregulating patched1 (*Dessaud et al., 2007*), and downstream targets regulate each other to generate discrete domains of expression (*Balaskas et al., 2012*). Thus, in contrast to the Nodal system which rapidly establishes target gene expression patterns, the Shh system makes extensive use of feedback inhibition and cross-regulation. At the other extreme, the Bicoid morphogen has already formed a quasi steady-state gradient before its target genes can be activated during zygotic genome activation (*Driever and Nüsslein-Volhard, 1988*; *Gregor et al., 2007*; *Porcher et al., 2010*). Bicoid concentration and affinity to regulatory chromatin elements are important (but not the sole [*Ochoa-Espinosa et al., 2009*; *Chen et al., 2012*]) determinants of target gene expression along the anterior-posterior axis of the *Drosophila* embryo (*Driever et al., 1989*; *Burz et al., 1998*). Thus, in contrast to the Nodal morphogen gradient, which evolves and is interpreted continuously under pre-steady state conditions, the Bicoid morphogen system makes only limited use of temporal strategies to modulate target gene response.

The influence of transcription rates and delays in morphogen interpretation raises the question how these processes might be regulated at the molecular level. Transcriptional delay might be achieved by a co-activator for target gene induction. Alternatively, a repressor might have to be eliminated for a target gene to become competent to respond. Transcription rates might be influenced by local chromatin structure, promoter strength, and by co-activators that boost or repressors that dampen the levels of target gene expression (*Li et al., 2007*; *Lupien et al., 2008*; *Hager et al., 2009*; *Kanodia et al., 2012*; *Peterson et al., 2012*; *Coulon et al., 2013*; *Oosterveen et al., 2013*; *Foo et al., 2014*; *Xu et al., 2014b*). In either case, our study suggests that the intensity and onset of target gene transcription can be major determinants in shaping morphogen gradient interpretation. Similar mechanisms might modulate other rapid and dynamic pattern formation processes (*Bolouri and Davidson, 2003*; *Lewis, 2003*; *de-Leon and Davidson, 2010*; *Oates et al., 2012*).

## Materials and methods

### Fish strains and transgenics

Fish were raised and maintained under standard conditions. Wild-type embryos were collected from TLAB in-crosses. MZoep*tz57* embryos were obtained as previously described (*Zhang et al., 1998*; *Gritsman et al., 1999*). Mutations in the *smad2* gene (ENSDARG0000006389, zv9) were screened for in the sperm of ENU-treated males by TILLING with primers encompassing exons 9 and 10.

### Outer primers
OSm2F1: 5′-CAATGGAGATAAGCCTGTGGC.
OSm2R4: 5′-TCTGCAAATGTTTTAAGCACTATTTCAG.

### Inner primers

ISm2F2: 5′-CTGTAATTTTAATATATTCATTTTTGCTGGC.
ISm2R3: 5′-TGCATAAATCTAATTGGCATTTTTAGATAAACC.

A stop mutation was selected in exon 9 (arg[335] > Stop) and heterozygous fish were produced by in vitro fertilization. After at least three outcrosses of smad2[vu99]/+ fish with a TLAB strain, smad2[vu99]/smad2[vu99] germline carrier fish were obtained by the germline transplantation technique as described (Ciruna et al., 2002). The carrier fish were in-crossed to produce MZsmad2 embryos. gfp-smad2 and h2b-rfp transgenic strains were generated by the meganuclease I-SceI technique as described (Thermes et al., 2002). egfp was introduced in frame upstream of the zebrafish smad2 coding sequence. Both gfp-smad2 and h2b-rfp were inserted downstream of a 5.3 kb fragment of the zebrafish β-actin promoter (gift of F Maderspacher) and the cassettes were subcloned into the I-SceI vector. GFP-Smad2 and H2B-RFP transgenic fish were intercrossed to generate double trans-heterozygotes.

## Embryo manipulations

### mRNA injections

Constructs were cloned in pCS2+ and mRNA for smad2, gfp-smad2, squint and activin were synthesized using the mMessage mMachine kit (Ambion, Grand Island, NY). cyclops and squint morpholinos (MOs) (Gene Tools, Philomath, OR) were previously described (Feldman and Stemple, 2001; Karlen and Rebagliati, 2001). Dechorionated embryos were injected at the one-cell stage with 0.5–1 nl of mRNA at the appropriate concentration.

### Transplantations

Cell transplantations were performed by mouth pipetting. 20–50 cells were transplanted from a donor (gfp-smad2/h2b-rfp or wild-type) to a host embryo (wild-type or squint injected MZoep). To test Nodal signaling maintenance after input removal, marginal cells of gfp-smad2/h2b-rfp or wild-type donor embryos at 30% epiboly were transplanted into the animal pole of stage-matched wild-type host embryos. Transplanted embryos were further incubated and processed for either confocal microscopy or for in situ hybridization. To analyze the relationships between activated Smad2 levels and Nodal target gene expression, animal pole cells of gfp-smad2/h2b-rfp embryos at 3.5 hpf or 4.5 hpf were transplanted into the animal pole of squint injected stage-matched MZoep embryos. Under these conditions, only donor cells in transplanted embryos can respond to Nodal. Transplanted embryos were incubated for 1 or 2 hr and processed for confocal microscopy and for in situ hybridization.

### Nodal induction dynamics

Wild-type embryos were first injected at the one-cell stage with 1 nl of cyclops and squint MOs mixture (at 0.2 mM and 4 µg/µl, respectively) to inhibit endogenous Nodal signals. Morphants were further injected with 0.5–1.5 nl of recombinant mouse Nodal protein (rmNodal, R&D Systems, Minneapolis, MN) at different concentrations in the extracellular space at 4 hpf (sphere stage). To test transcriptional competence, rmNODAL was injected at different stages ranging from 3.25 to 5.25 hpf. Cycloheximide (Sigma, Saint Louis, MO) was applied to dechorionated embryos in embryo medium at 50 µg/µl at 3 hpf. 30 min later embryos were injected with 100 nM Nodal protein and were incubated for 90 min before being processed for total RNA extraction and NanoString analysis.

## Embryo samples processing

### In situ hybridization

In situ hybridization on whole mount embryos were carried out using standard protocols. ntl, gsc, foxa3, efnb2a, efnb2b and bra probes were previously described (Schulte-Merker et al., 1992, 1994; Bennett et al., 2007; Martin and Kimelman, 2008). A ntl-gsc fusion probe was constructed by amplifying the full length gsc mRNA using the primers cg [ggatcc] ATGCCCGCTGGGATGTTTAGTATC and ataagaat [gcggccgc] TTAGATATTACTTTAATATTTGTTCCTGTTTTCAGGC and cloning into a plasmid containing full length ntl mRNA using BamHI and NotI enzymes. This construct was linearized with NotI and transcribed using T3. The mRNA was injected into embryos collected from a TLAB incross at the 1-cell stage at four different doses (2 pg, 8 pg, 25 pg, and 100 pg). Embryos were cultured to the 128–256-cell stage and fixed in 4% formaldehyde overnight. They were dehydrated, rehydrated, and stained using standard methods. The same concentrated probe stocks were used that had

been used in *Figure 2*, which were freshly diluted 1:100 in hybridization medium. Following staining, embryos were cleared in benzyl benzoate/benzyl alcohol (2:1 vol/vol) Five random embryos were chosen for each probe–dose combination and imaged from the animal cap in one session with no changes to the settings of the microscope. These images were blinded (so that the file names no longer reflected the treatment), converted to grayscale, and the region of the animal cap that represented a single layer of stained cells was selected in ImageJ with the Lasso tool. The mean brightness of this region and a similarly sized background region were calculated. Each image file was quantitated three separate times and averaged, and the background brightness was subtracted. Brightness measurements were inverted (so that more staining would be a higher number), and their mean and standard deviation was plotted.

## Western blotting

Western blots were performed using standard procedures and the signal was detected by chemiluminescence (ECL plus, Amersham, Piscataway, NJ). Phosphorylated Smad2 was probed using a rabbit anti-phospho-Smad2 (Ser465/467) antibody (1:2000 dilution, Cell Signaling Technology, Danvers, MA, #3104) and total Smad2 was probed using a rabbit anti-Smad2/3 antibody (1:2000 dilution, Cell Signaling Technology, #3102). Signals were quantified in ImageJ with the Gel Analyzer function and the ratio between phospho-Smad2 and total Smad2 signals was calculated.

## NanoString

Total RNA from 5 to 10 embryos for each data point was extracted using the RNAeasy mini kit (Qiagen, The Netherlands) and 100 ng of input RNA was processed through the nCounter assay using standard protocols (NanoString Technologies, Seattle, WA) (*Kulkarni, 2011*). Samples were first normalized to positive controls included in the codeset. The codeset content was further normalized to 11 reference genes to correct for difference in sample input between assays, according to manufacturer's guidelines.

## qPCR

Total RNA from 5 to 10 embryos was extracted using the RNAeasy mini kit (Qiagen) and 100 ng of input RNA was used to synthesize cDNA with the iScript kit (Bio-Rad, Hercules, CA). qPCR reactions were performed in duplicates using the Go Taq qPCR kit (Amersham) on a MX3000P qPCR instrument (Agilent Technologies, Santa Clara, CA). Relative expression of a given gene was calculated by the ΔCt procedure using e*ef1a1l1* as a reference. Primers used for qPCR analysis are as followed:

*bra*
F: 5′ CTGTAGGGAACTCCTCTCAGT
R: 5′ AAGCAGCTGTGTCGTATAAAG
*eef1a1l1*
F: 5′ AGAAGGAAGCCGCTGAGATGG
R: 5′ TCCGTTCTTGGAGATACCAGCC
*flh*
F: 5′ GGCGGAGATGAGAGAACGAAC
R: 5′ GATAGCAGAACACGGGATAGC
*gsc*
F: 5′ GAGACGACACCGAACCATTT
R: 5′ CCTCTGACGACGACCTTTTC

## Time lapse imaging, image processing, analysis and cell tracking

Live embryos were embedded in 0.8% low melting point agarose on a glass bottom culture dish (MatTek, Ashland, MA), with the marginal region facing the objective. The dish was filled with fish water (Instant Ocean sea salt [0.6 g/l] in RO water, 0.01 mg/l methylene blue) to prevent dehydration. Images were acquired on a PASCAL confocal microscope (Zeiss, Germany) using a 25× objective (LCI Plan-Neofluar/0.8) equipped with a heated stage set at 28°C. Samples were simultaneously excited with an argon laser at 488 nm and a Helium laser at 546 nm. Four confocal planes were imaged at 3 μm intervals (512 × 512 size, 12-bit depth, line averaging eight times) every 3 min for a period of 3 hr. Embryonic position of the recorded field was assessed morphologically at the end of the imaging session. Image stacks were processed using custom-made Matlab scripts to measure centroid localization of nucleus, nucleo-cytoplasmic ratio of GFP-Smad2 intensity, distance from the margin,

and cell tracks (see *Supplementary file 1*). Channels of stacked confocal images were split in ImageJ and saved as grayscale TIFF image sequences (8-bit for H2B-RFP, 16-bit for GFP-Smad2). H2B-RFP images were further converted to binary images, by applying a threshold using Otsu's method. Objects smaller than 20 pixels were then removed, and the resulting images were segmented using the Moore-Neighbor tracing algorithm modified by Jacob's stopping criteria. The centroid location and area of each nucleus were then extracted. The binary image of the H2B-RFP was used as a mask on the corresponding GFP-Smad2 image to extract nuclear only- and cytoplasmic only GFP-Smad2 signals. The ratio between the mean nuclear GFP intensity and the mean cytoplasmic GFP intensity was used to define Smad2 activity at the single cell level. In MZ*oep* mutants (Nodal insensitive), the mean NC ratio value is 1.19 ± 0.07. Cell tracking was perfomed using the nearest-neighbor strategy based on the centroid position of each nucleus at different time frames. The nearest centroid of the next frame was selected as being part of the cell track if it was less than 10 pixels apart. This process was reiterated through all the frames to generate cell tracks. Based on visual checks of the resulting tracks, ∼90% of the tracks are estimated to be accurate. The distance between each centroid and the margin was measured at each time point. The position of the margin was defined using a user interface: the maximal projection of the H2B-RFP channel was displayed and six reference points were manually selected along the yolk-blastoderm boundary. The whole margin position was then extrapolated by fitting a polynomial curve. The fitted function was used to determine the distance of each centroid from the margin.

## Smad2/FoxH1 chromatin immuno-precipitation

Embryos for Smad2 and FoxH1 ChIP were collected at dome stage after 5 pg *squint* mRNA injection or after treatment with the Nodal signaling inhibitor SB505124 (Sigma S4696) at 20 μM final. For FoxH1 ChIP, embryos were injected with 5 pg of *FoxH1-flag* mRNA at 1-cell stage, and anti-flag antibody was used for the pull down.

For each ChIP, 800 embryos were collected and fixed in 1.85% formaldehyde for 15 min at 20°C. Formaldehyde was quenched by adding glycine to a final concentration of 0.125 M. Embryos were rinsed three times in ice-cold PBS, and resuspended in cell lysis buffer (10 mM Tris-HCl pH7.5/10 mM NaCl/0.5% NP40) and lysed for 15 min on ice. Nuclei were collected by centrifugation, resuspended in nuclei lysis buffer (50 mM Tris-HCl pH 7.5/10 mM EDTA/1% SDS) and lysed for 10 min on ice. Samples were diluted three times in IP dilution buffer (16.7 mM Tris-HCl pH 7.5/167 mM NaCl/1.2 mM EDTA/ 0.01% SDS) and sonicated to obtain fragments of ∼500 bp. Triton X-100 was added to a final concentration of 0.75% and the lysate was incubated overnight while rotating at 4°C with 25 μl of protein G magnetic Dynabeads (Invitrogen) pre-bound to an excess amount of antibody. Antibodies used were anti-FLAG M1 (Sigma F3165), anti-Smad2/3 (Invitrogen, Grand Island, NY 51–1300). Bound complexes were washed six times with RIPA (50 mM HEPES pH7.6/1 mM EDTA/0.7% DOC/1% Igepal/0.5 M LiCl) and TBS and then eluted from the beads with elution buffer (50 mM NaHCO3/1% SDS). Crosslinks were reversed overnight at 65°C and DNA purified by the QIAquick PCR purification kit (Qiagen). Libraries were prepared according to the Illumina sequencing library preparation protocol and sequenced on an Illumina HiSeq 2000. ChIP-seq reads were mapped to the zebrafish genome (UCSC Zv9 assembly) and peaks were called using MACS (*Zhang et al., 2008*).

## Nodal signaling modeling

The goal of modeling Nodal signaling is to predict the range of expression of Nodal target genes in the embryonic blastula and to analyze the key parameters regulating gene response.

*Equations 1–3* were used to model the kinetics of Nodal signaling.

$$\frac{\partial N}{\partial t} = P(x,t) + D_N.\nabla^2.N - k_1.N,$$

$$\frac{dS_p}{dt} = k_2.N.S - k_3.S_p,$$

$$\frac{dRNA_{target}}{dt} = \alpha.\frac{S_p^n}{K_d^n + S_p^n} - \beta.RNA_{target}.$$

P: Production rate of Nodal from the source, where $P = \gamma.\frac{t}{1+t}$ when x ≤ 25 μm and $P = 0$ when x > 25 μm.

$D_N$: Diffusion coefficient of Nodal.
$k_1$: clearance rate of Nodal.
$k_2$: activation (phosphorylation) rate of Smad2.
$k_3$: de-activation (de-phosphorylation) rate of Smad2.
$\alpha$: maximal transcription rate of Nodal target gene.
$\beta$: degradation rate of Nodal target gene.
$K_d$: effective dissociation constant of activated Smad2 for target gene enhancer.
$n$: Hill coefficient.
We assume that the pool of total Smad2 remains constant such that $S_{total} = S + S_p$.

We used two different scenarios to reflect the experimental set up: the 'homogenous' scenario, where ectopic Nodal ligand is injected uniformly into a Nodal depleted embryo, and the 'spatial gradient' scenario, where Nodal is produced locally on one side of a one-dimension column of cells.

## Parameterization
Known parameters: $D_N$: 1.5 µm²/s; $k_1$: 1 × 10⁻⁴ s⁻¹ (**Müller et al., 2012**); Estimated parameters: Smad2$_{total}$: 25 nM/cell (**Schmierer et al., 2008**); Unknown parameters: $\gamma$, $k_2$, $k_3$, $\alpha$, $\beta$, $K_d$, $n$.

## Homogenous model
In this model, we assume that the exogenous Nodal concentration is uniformly distributed in the embryo and reaches steady-state shortly after injection. In this case, $\frac{\partial N}{\partial t} = 0$;
Initial conditions:

$$N = 0, 5, 10, 25, 50 \text{ or } 100 \text{ nM},$$

$$RNA_{target} = 0.$$

Unknown parameters were then retrieved through minimization of the residual sum of squared errors for the fitted model using the Nelder-Mead simplex method (using a constrained version of the MATLAB function *fminsearch*) using three different initial guesses spanning the parameter space. The best set of parameters was selected according to the highest coefficient of determination. Parameter confidence intervals of 95% were computed from the residuals and the coefficient covariance.

## Spatial gradient model
In this model, we consider the behavior of a column of cells spanning the vegetal–animal axis of the embryo (500 µm in length) during a 3-hr time span. Parameter values for Smad2 activation and gene induction were taken from the homogenous model. The unknown parameter left in this spatial model is $\gamma$, the maximal production of Nodal. As expected for a morphogen molecule, the system is very sensitive to the levels of Nodal. We thus manually set $\gamma$ to a value where the simulated expression pattern of the well-characterized Nodal target *ntl* fits the in vivo distribution ($\gamma = 0.03$ nM/s). Each gene was then individually simulated using its specific parameters, and its range of expression was defined as the distance from the source where the expression drops below 100 counts.
Initial conditions:

$$N = 0; S = S_{tot}; RNA_{target} = 0;$$

## Boundary conditions

$$\frac{\partial N}{\partial x}\bigg|_{x=0} = \frac{\partial N}{\partial x}\bigg|_{x=500} = 0.$$

Simulations were solved numerically using the MATLAB *pdepe* function.

## Delay model
In order to model the delay in gene induction, we introduced a co-factor *Y* required to activate target gene transcription in cooperation with Smad2. In this case,

$$\frac{dRNA_{target}}{dt} = \alpha \cdot \frac{Y \cdot S_p}{K_d^2 + Y \cdot S_p} - \beta \cdot RNA_{target},$$

and

$$\frac{dY}{dt} = k_4, \tag{4}$$

with $k_4 = 4.6 \times 10^{-5}$ nM/s and with initial conditions at $T_0 = 0$, $Y_0 = 0$. To abolish delay, we solved (3) with initial conditions where at $T_0$, $Y_0 = Y_{final}$.

## Model comparison and complexity

Most models of morphogen signaling and interpretation use large numbers of parameters and can thus suffer from overfitting. We thus considered six different models with different numbers of parameters to describe Smad2-dependent transcription of Nodal target genes and compared the probability that these models can generate the data.

The same rate of Smad2 activation is shared among these models:

$$\frac{dS_p}{dt} = k_1.N.S - k_2.S_p,$$

where $S_p$, $N$ and $S$ are phosphorylated Smad2, Nodal and non-phosphorylated Smad2 concentrations, respectively, with $k_1 = 3.1 \times 10^{-6}$ nM$^{-1}$s$^{-1}$ and $k_2 = 1.8 \times 10^{-4}$ s$^{-1}$.

All models for RNA production have two terms: a pSmad2-dependent mRNA transcription rate, and a linear mRNA degradation rate.

In Model 1, we assume that the effective transcription rate is linearly proportional to pSmad2 concentration:

$$\text{M1:} \quad \frac{dRNA}{dt} = \alpha.S_p - \beta.RNA.$$

In Model 2, we assume that the transcription rate is regulated by a dissociation constant for pSmad2:

$$\text{M2:} \quad \frac{dRNA}{dt} = \frac{S_p}{K_d + S_p} - \beta.RNA.$$

Model 3 is similar to Model 2, with the addition of a maximum transcription rate coefficient:

$$\text{M3:} \quad \frac{dRNA}{dt} = \alpha.\frac{S_p}{K_d + S_p} - \beta.RNA.$$

Model 4 is similar to Model 2, with the addition of a Hill coefficient and a fixed transcription rate coefficient.

$$\text{M4:} \quad \frac{dRNA}{dt} = A.\frac{S_p^n}{K_d^n + S_p^n} - \beta.RNA,$$

A = 0.1 count/s, corresponding to the mode value of the maximal transcription rate coefficient distribution of our fully developed model (see *Table 1*).

Model 5 is similar to Model 4, except that in this case, the maximal transcription rate coefficient is let free while the dissociation constant is fixed:

$$\text{M5:} \quad \frac{dRNA}{dt} = \alpha.\frac{S_p^n}{C^n + S_p^n} - \beta.RNA,$$

C = 6.7 nM, corresponding to the mode value of the dissociation constant distribution of our fully developed model (see *Table 1*).

Finally, Model 6 is the fully developed model:

$$\text{M6:} \quad \frac{dRNA}{dt} = \alpha.\frac{S_p^n}{K_d^n + S_p^n} - \beta.RNA.$$

Assuming that our NanoString measurements $\{y_i\}$ are noisy with a standard deviation of $\{\sigma_i\}$, we can consider $y_i$ as a Gaussian random variable with a mean value $f(t_i;\theta)$ of the underlying model containing a vector of parameters $\theta$ and a variance $\sigma_i^2$ (*Bialek, 2012*). We thus have,

$$P(y_i|t_i,\theta) = \frac{1}{\sqrt{2\pi\sigma_i^2}}\exp\left[-\frac{(y_i - f(t_i;\theta))^2}{2\sigma_i^2}\right],$$

and

$$P(\{y_i\}|\{t_i\},\theta) = \prod_{i=1}^{N} P(y_i|t_i,\theta).$$

The probability of the data given the underlying model is

$$P\left(\{t_i, y_i\}|\theta\right) = \left[\prod_{i=1}^{N} P(y_i|t_i,\theta)\right]\left[\prod_i P(t_i)\right].$$

Given

$$\chi^2 = \sum_i \left|\frac{(y_i - f(t_i;\theta))}{\sigma_i}\right|^2,$$

$$P\left(\{t_i, y_i\}|\theta\right) = \exp\left[\sum_{i=1}^{N} \ln P(t_i) - \frac{1}{2}\sum_{i=1}^{N} \ln\left(2\pi\sigma_i^2\right) - \frac{1}{2}\chi^2\right].$$

Therefore minimizing $\chi^2$ by fitting the parameters $\theta$ increases the probability that the model could have produced the data. However, different classes of models with different numbers of parameters whose values are unknown have to be considered. To determine the probability of the data given a class of models with unknown K parameters, an integration over all the possible values of the parameters, weighted by some prior knowledge, has to be computed

$$P(\{t_i, y_i\}|class) = \int d^K\theta P(\theta)P(\{t_i, y_i\}|\theta)$$

$$= \int d^K\theta P(\theta)\exp\left[-\frac{1}{2}\sum_{i=1}^{N} \ln\left(2\pi\sigma_i^2\right) - \frac{1}{2}\chi^2\left(\theta; \{t_i, y_i\}\right)\right]\left[\prod_n P(t_n)\right],$$

where $P(\theta)$ is the probability of the a priori distribution of the parameters.

$\chi^2$ is proportional to N, and we can write

$$P(\{t_i, y_i\}|class) = \exp\left[-\frac{1}{2}\sum_{i=1}^{N} \ln\left(2\pi\sigma_i^2\right)\right]\left[\prod_n P(t_n)\right]\int d^K\theta e^{-Nf(\theta)},$$

where

$$f(\theta) = \frac{1}{2N}\chi^2(\theta; \{t_i; y_i\}) - \frac{1}{N}\ln P(\theta).$$

We use a saddle point approximation such that

$$\int d^K\theta e^{-Nf(\theta)} \approx e^{-Nf(\theta^*)}(2\pi)^{K/2}\exp\left[-\frac{1}{2}\ln \det(N\mathscr{H})\right],$$

where $\theta^*$ is the value at which $f(\theta)$ is minimized, and H is the Hessian matrix of the second derivatives at this point. Taking the negative log probability of the data given the model class, we have

$$-\ln P(\{t_i, y_i\}|class) \approx \sum_{i=1}^{N} \ln\left(2\pi\sigma_i^2\right) - \sum_{i=1}^{N} \ln P(t_i) + \frac{1}{2}\chi^2_{min} + \ln P(\theta^*) - \frac{K}{2}\ln 2\pi + \frac{1}{2}\ln \det(N\mathscr{H}).$$

Since H is a K × K matrix, $\det(N\mathscr{H}) = N^K\det(\mathscr{H})$, and we finally have

$$-\ln P(\{t_i, y_i\}|class) \approx \sum_{i=1}^{N} \ln\left(2\pi\sigma_i^2\right) - \sum_{i=1}^{N} \ln P(t_i) + \frac{1}{2}\chi^2_{min} + \frac{K}{2}\ln N + \frac{1}{2}\ln \det(\mathscr{H}) + \ln P(\theta^*) - \frac{K}{2}\ln 2\pi.$$

The negative log probability measures the length of the shortest code for the data being generated given the class of models. This length depends on the sample size of the data, the number of parameters, the quality of the fit, and some prior on the parameters that we consider flat in our case. Therefore, the model giving the smallest value of the code is to be considered the best model explaining the data given the sample size. The NanoString data and associated noise (which is ~10% of the count value based on the analysis of the positive spikes across a cartridge) are identical in all our models, so we are left to compare

$$C_{MX} = \frac{1}{2}\chi_{min}^2 + \frac{K}{2}\ln N + \frac{1}{2}\ln \det(\mathscr{H}) - \frac{K}{2}\ln 2\pi.$$

Calculating the mean value across all genes, we found
$C_{M1} = 601.1$
$C_{M2} = 767.6$
$C_{M3} = 681.0$
$C_{M4} = 264.3$
**$C_{M5} = 134.5$**
$C_{M6} = 204.1$
$C_{M5} < C_{M6} < C_{M4} < C_{M1} < C_{M3} < C_{M2}$.

Thus, among all the six different models we considered, model M5 and M6 are the most probable models given the data, highlighting the importance of the maximal transcription rate.

## Acknowledgements

The authors thank Bérengère Dubrulle-Bréon for suggestions on modeling, Sharad Ramanathan for help with cell tracking algorithms, modeling and statistical analysis, Will Talbot, Susan Mango, Sharad Ramanathan, Nathan Lord and members of the Schier lab for critical reading of the manuscript. This work was supported by NIH grants to LS-K and AFS.

## Additional information

### Funding

| Funder | Author |
| --- | --- |
| National Institutes of Health (NIH) | Alexander F Schier, Lilianna Solnica-Krezel |

The funder had no role in study design, data collection and interpretation, or the decision to submit the work for publication.

### Author contributions

JD, Conception and design, Acquisition of data, Analysis and interpretation of data, Drafting or revising the article; BMJ, Analysis and interpretation of data; LA, JAF, Acquisition of data, Analysis and interpretation of data; S-HK, LS-K, Contributed unpublished essential data or reagents; AFS, Conception and design, Analysis and interpretation of data, Drafting or revising the article

### Ethics

Animal experimentation: This study was performed in strict accordance with the recommendations in the Guide for the Care and Use of Laboratory Animals of the National Institutes of Health. All of the animals were handled according to approved institutional animal care and use committee (IACUC) protocols (#25-08) of Harvard University.

## Additional files

### Supplementary files

• Supplementary file 1. Scripts used for tracking cells and measuring the nucleo-cytoplasmic ratio.

• Source code 1. (runPdeSysTestSourceNoLefty.m): Simulation of the spatial distribution of Nodal, acitve Smad2 and Nodal target gene.

• Source code 2.   (runOdeSysplotNanoM1.m): Simulation of the time- and Nodal concentration-dependent induction of target genes.

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
