## [Decision Letter]

Thank you for sending your work entitled “Response to Nodal morphogen gradient is determined by the kinetics of target gene induction” for consideration at *eLife*. Your article has been favorably evaluated by Janet Rossant (Senior editor), a Reviewing editor, and 2 reviewers.

The Reviewing editor and the reviewers discussed their comments before we reached this decision, and the Reviewing editor has assembled the following comments to help you prepare a revised submission.

There is uniform consensus that your study addressing the mechanisms transforming the graded distribution of Nodal into distinct RNA expression-pattern domains is solid and important. This topic is central in Developmental Biology and is highly relevant for a large number of other processes during later developmental stages in which Nodal and other secreted TGF-b molecules are involved. The subject and scope of the results reported in the paper thus fit very well the requirements from papers published in *eLife*.

Please address the following specific points (brought in order of appearance in the text; points 5 and 6 can be viewed as the major concerns).

1) While convincing, the rescue of the MZ*smad* mutation by GFP-Smad2 appears incomplete as compared with the wild type and to the rescue by non-GFP version. If this really reflects the functionality of the protein, the authors should mention this point in the text.

2) “Cells close to the margin tended to activate Smad2 early and reached the highest levels of activated Smad2” (mentioned in the Results). In Figure 2, the second and fourth cells from the margin do not appear to follow this rule. If they are exceptions, they should be replaced. Alternatively, instead of following one cell per distance, presenting the average of values measured from several adjacent cells in 50 micrometer intervals would probably provide a more convincing picture of what appears to be a correct conclusion.

3) Figure 2: the time interval between the 5 frames presented should be indicated in the figure legend. Additionally, for the panel to be informative, at least for several representative cells, the direction of the movement should be indicated.

4) “Cells located farther away from the margin tended to activate Smad2 with a delay and the levels of activated Smad2 remained low” (in the Results). In Figure 2, it appears that at 3.5 hpf, Smad2 is activated in cells away from the margin (80 µM and further). Subsequently, this activation is gradually lost (what appears like inactivation from 4 to 5 hpf). This is also seen in some examples shown in Figure 2 as mentioned in point 3. The '3.5 hpf results' can be interpreted as either 'noise' or as a process of inactivation with time of cells away from the source. The latter explanation (activation followed by inactivation) could be discussed in the context of the ratchet model. If the authors consider the activation values at 3.5 hpf to signify noise, they should clearly state that in the text and indicate accordingly the range of relevant data Figure 2 (starting from 4 hpf?).

5) “*Ntl* was first faintly detected in a few cells on the presumptive dorsal side of the embryo at the midblastula stage. Subsequently, its expression domain intensified and progressively extended animally until the onset of gastrulation. By contrast, *gsc* expression initiated ∼30 min later and remained confined to a narrow domain on the dorsal side”. As shown in Figure 2, the authors use in situ hybridization to compare the dynamics of transcription and spatial distribution of *ntl* and *gsc* transcripts. For the conclusion to be valid, the authors should show that the sensitivity of the method is identical for both genes. This could be done for example by injecting a concentration series of a *gsc-ntl* fused RNA, detecting it at 3 hpf (prior to endogenous gene activation) using *ntl* and *gsc* probes (used in the study) to detect the injected RNAs. Comparable sensitivity of both probes should be confirmed. Further, identical staining procedure (e.g., duration of color development) should be applied to both probes to validate the conclusion as above. While I believe the result will be the expected one (especially in light of the nanostring data) formally, based on this panel alone and the description of the procedure, the difference in timing and expression pattern could arise from relatively more efficient *ntl* detection. While not essential, double labeling within the same embryo would be even more convincing.

6) Concerning Figure 3, the labeling of the host embryo as MZ*oep* + Nodal should be changed into MZ*oep* + nodal to indicated that the source of the Nodal is RNA. In that figure, GFP-Smad2 NC values at 5.5 hpf are interpreted by the authors to signify similar activation levels in the transplanted cells. It is possible, that cells transplanted into 3.5hpf embryos vs those transplanted into 4.5hpf embryos, could have experienced different levels of Nodal activation in the host. For example, it could in principle be that the Nodal level the transplanted cells are exposed to is higher in the “3.5hpf experiment” due to the protein expression dynamics derived from the injected RNA. To rule out this possibility, the authors could determine the N/C ratio of GFP-Smad2 at 4, 4.5 and 5 hpf as well. This experiment is suggested despite the statement that the cells do not remember the signal when transplanted into animal positions (as in Figure 3—figure supplement 1) since high initial signaling could in principle cooperate with a later weaker signal.

7) Figure 3—figure supplement 1, statistical tests showing significance for panel A would make the statement more convincing.

8) The authors state: “These analyses indicate that not only the K_d_ but also maximal transcription rate is a key determinant of gene-specific morphogen response...” From Figure 7, the K_d_ appears to have no effect on either expression rate or expression range. It is not clear therefore why the authors consider the K_d_ as a contributing parameter.

9) In Figure 8, bottom right panel, are the counts significantly higher following the earlier injection at the same developmental stage as those counts following a later injection? If so, the author should discuss this point. If the data exist, it would be interesting to discuss this point in the context of the genes analyzed in Figure 8—figure supplement 1 as shown in the right panel.

---

## [Author Response]

*1) While convincing, the rescue of the MZ*smad *mutation by GFP-Smad2 appears incomplete as compared with the wild type and to the rescue by non-GFP version. If this really reflects the functionality of the protein, the authors should mention this point in the text.*

In the revised manuscript we added a supplementary table that shows that the larval lethality of Z*smad2* mutants can be rescued to adulthood by the GFP-Smad2 transgene used throughout the study and clarify that *gfp-smad2* mRNA can rescue MZ*smad2* mutants but not as effectively as *smad2* mRNA.

*2)* “*Cells close to the margin tended to activate Smad2 early and reached the highest levels of activated Smad2*” *(mentioned in the Results). In*
Figure 2*, the second and fourth cells from the margin do not appear to follow this rule. If they are exceptions, they should be replaced. Alternatively, instead of following one cell per distance, presenting the average of values measured from several adjacent cells in 50 micrometer intervals would probably provide a more convincing picture of what appears to be a correct conclusion*.

We added additional cell tracks to show the general trend of Smad2 activation. We added the following text to the legend of Figure 2: “The short bursts observed in some cell tracks are caused by transient nuclear accumulation of GFP-Smad2 at the onset of nuclear envelope breakdown and are observed even in the absence of Nodal signaling.”

*3)*
Figure 2*: the time interval between the 5 frames presented should be indicated in the figure legend. Additionally, for the panel to be informative, at least for several representative cells, the direction of the movement should be indicated*.

We added this information in the legend of Figure 2. All cells move towards the vegetal pole because of epiboly movements (shown in Figure 2—figure supplement 2), but maintain their position with respect to the margin, as shown in Figure 2.

*4)* “*Cells located farther away from the margin tended to activate Smad2 with a delay and the levels of activated Smad2 remained low*” *(in the Results). In*
Figure 2*, it appears that at 3.5 hpf, Smad2 is activated in cells away from the margin (80 µM and further). Subsequently, this activation is gradually lost (what appears like inactivation from 4 to 5 hpf). This is also seen in some examples shown in*
Figure 2
*as mentioned in point 3. The '3.5 hpf results'* can *be interpreted as either 'noise' or as a process of inactivation with time of cells away from the source. The latter explanation (activation followed by inactivation) could be discussed in the context of the ratchet model. If the authors consider the activation values at 3.5 hpf to signify noise, they should clearly state that in the text and indicate accordingly the range of relevant data*
Figure 2
*(starting from 4 hpf?)*.

We added the following information to the legend of Figure 2: “Basal NC ratio is higher in younger embryos (see Figure 2 3.5 hpf). Since this phenomenon is also observed in the absence of Nodal signaling (MZ*oep* mutants), the higher NC ratio is unlikely to reflect early Smad2 activation, but rather a higher nuclear import/export ratio of GFP-Smad2 during early development.”

*5)* “Ntl *was first faintly detected in a few cells on the presumptive dorsal side of the embryo at the midblastula stage. Subsequently, its expression domain intensified and progressively extended animally until the onset of gastrulation. By contrast,* gsc *expression initiated ∼30 min later and remained confined to a narrow domain on the dorsal side*”*. As shown in*
Figure 2*, the authors use* in situ *hybridization to compare the dynamics of transcription and spatial distribution of* ntl *and* gsc *transcripts. For the conclusion to be valid, the authors should show that the sensitivity of the method is identical for both genes. This could be done for example by injecting a concentration series of a* gsc-ntl *fused RNA, detecting it at 3 hpf (prior to endogenous gene activation) using* ntl *and* gsc *probes (used in the study) to detect the injected RNAs. Comparable sensitivity of both probes should be confirmed. Further, identical staining procedure (e.g., duration of color development) should be applied to both probes to validate the conclusion as above. While I believe the result will be the expected one (especially in light of the nanostring data) formally, based on this panel alone and the description of the procedure, the difference in timing and expression pattern could arise from relatively more efficient* ntl *detection. While not essential, double labeling within the same embryo would be even more convincing.*

We performed the experiment suggested by the reviewer and present the results in Figure 2—figure supplement 3. We found that the *ntl* probe is slightly better than the *gsc* probe but to such a small extent that it is unlikely to dramatically skew the detection of endogenous gene expression. As an additional quantitative test for *gsc* and *ntl* expression, we analyzed recent single-cell transcriptome data (Satija et al. Nature Biotechnology, in press). In this paper, we and the Regev lab mapped the expression of the zebrafish transcriptome to different bins in the 50% epiboly embryo (late blastula). As shown in Figure 2—figure supplement 4, *ntl* is expressed in a broader domain and at higher levels than *gsc*. This conclusion is also supported by the nanostring data and numerous papers that analyzed *gsc* and *ntl* expression in zebrafish and *Xenopus* (see for example Füller et al., Dev. Biol. 2014; Eimon & Harland, Development 2002, Bell et al., Development 2003, Webster et al., BMC Dev. Biol. 2009).

*6) Concerning*
Figure 3*, the labeling of the host embryo as MZ*oep *+ Nodal should be changed into MZ*oep *+ nodal to indicated that the source of the Nodal is RNA. In that figure, GFP-Smad2 NC values at 5.5 hpf are interpreted by the authors to signify similar activation levels in the transplanted cells. It is possible, that cells transplanted into 3.5hpf embryos vs. those transplanted into 4.5hpf embryos, could have experienced different levels of Nodal activation in the host. For example, it could in principle be that the Nodal level the transplanted cells are exposed to is higher in the* “*3.5hpf experiment*” *due to the protein expression dynamics derived from the injected RNA. To rule out this possibility, the authors could determine the N/C ratio of GFP-Smad2 at 4, 4.5 and 5 hpf as well. This experiment is suggested despite the statement that the cells do not remember the signal when transplanted into animal positions (as in*
Figure 3—figure supplement 1*) since high initial signaling could in principle cooperate with a later weaker signal*.

This comment implies that cells in the 3.5 hpf experiment might have been exposed to Nodal not only for a longer time but experienced a peak of pathway activation before 5.5 hpf. As proposed by the reviewer, we measured GFP-Smad2 levels in a time course experiment and found that the NC ratio of GFP-Smad2 is not higher at any time between 3.5 and 5.5 hpf as compared to 5.5 hpf. This data is now shown in Figure 3—figure supplement 1.

*7)*
Figure 3—figure supplement 1*, statistical tests showing significance for panel A would make the statement more convincing*.

Added in revised manuscript.

*8) The authors state:* “*These analyses indicate that not only the K*_*d*_
*but also maximal transcription rate is a key determinant of gene-specific morphogen response...*” *From*
Figure 7*, the K*_*d*_
*appears to have no effect on either expression rate or expression range. It is not clear therefore why the authors consider the K*_*d*_
*as a contributing parameter*.

We revised the text as follows: “Although the K_d_ may affect target gene response, these analyses indicate that the maximal transcription rate is a key parameter in morphogen response.” We wish to be conservative in our claims and do now want to dismiss the importance of the K_d_ in morphogen interpretation.

*9) In*
Figure 8*, bottom right panel, are the counts significantly higher following the earlier injection at the same developmental stage as those counts following a later injection? If so, the author should discuss this point. If the data exist, it would be interesting to discuss this point in the context of the genes analyzed in*
Figure 8—figure supplement 1
*as shown in the right panel*.

The counts for late-expressed target genes are indeed higher following early injections. We added the following text to the legend of Figure 8: “Note that counts for the expression of late target genes are higher after early injection compared to later injections. This effect might be due to the fact that after early injections phospho-Smad2 levels are high for a longer period before a gene becomes competent to respond as compared to late injections, when there is a shorter time window of high phospho-Smad2 levels. There might be a priming mechanism in which longer exposure to activated Smad2 increases gene expression when competence is reached.”